# THE BLESSING OF SMOOTH INITIALIZATION 🌀 FOR VIDEO DIFFUSION MODELS

## ABSTRACT

Extending the success of text-to-image (T2I) synthesis to text-to-video (T2V) synthesis is a promising direction for visual generative AI. Popular training-free sampling algorithms currently generate high-fidelity images within the Stable Diffusion family. However, when applied to video diffusion models (VDMs), these techniques result in limited diversity and quality due to the low-quality data in video datasets. We focus on inference to mitigate this issue, and then we propose a training-free paradigm that optimizes the initial Gaussian noise by introducing a targeted semantic prior bias into the sampling process from a *smoothing* perspective. The paradigm significantly improves both the fidelity and semantic faithfulness of the synthesized videos. Guided by theoretical analysis using random *smoothing* and differential equations, our resulting method SmoothInit can be understood as approximately incorporating third-order derivatives into gradient descent, which contributes to be better convergence in learning semantic information. A more efficient version, Fast-SmoothInit , is proposed to achieve better experimental results by leveraging a momentum mechanism. Both SMOOTHINIT and FAST-SMOOTHINIT demonstrate promising empirical results across various benchmarks, including UCF-101/MSR-VTT-related FVD, Chronomagic-bench, and T2V-Compbench, setting a new standard for noise initialization in VDMs.

## 1 INTRODUCTION

Text-to-video (T2V) synthesis (Ho et al., 2022; Singer et al., 2022; Bao et al., 2024; Blattmann et al., 2023) is a promising topic in artificial intelligence-generated content (AIGC). In contrast to text-to-image (T2I) synthesis (Rombach et al., 2022; Esser et al., 2024; Peebles & Xie, 2023), which benefits from a large, well-curated dataset such as LAION-5B (Schuhmann et al., 2022) of image-text pairs to train powerful diffusion models for generating diverse and high-quality images, T2V generation methods often produce suboptimal results due to the cluttered, watermarked nature and small scale of existing training datasets (Bain et al., 2021; Soomro, 2012). Compared to retraining a desired video diffusion model (VDM), one of the most straightforward and inexpensive solutions is to develop ideal plug-and-play algorithms to enhance the fidelity and diversity of composite videos. Unfortunately, the training-free sampling algorithms (Song et al., 2023a; Lu et al., 2022c; Bao et al., 2022) that perform well in T2I synthesis often struggle to meet expectations in T2V generation[1]. Given this, while recent approaches (Wu et al., 2023; Qiu et al., 2024; Chen et al., 2024b; Jeong et al., 2024) leverage the sampling characteristic of VDMs to enhance the quality of video by ensuring temporal consistency and semantic faithfulness, there remains a lack of genuine attempts to comprehend and explore the training-free sampling paradigm from the perspective of initial noise.

In this paper, we aim to answer the question: "*To what extent can just optimizing the initial Gaussian noise improve the generation ability of VDMs?*". Sufficient empirical results (Qi et al., 2024; Ban et al., 2024; Mao et al., 2023; Shirakawa & Uchida, 2024) consistently demonstrate that the synthesized image is highly sensitive to the initial Gaussian noise, particularly when sampling through ordinary differential equation-based (ODE-based) sampling algorithms (Lu et al., 2022b; Song et al., 2023a). Even injecting a slight perturbation into the initial noise can significantly alter the generated object's color, position, and morphology. These observations suggest that there may exist a special

---

[1]We provide an obvious example in Appendix E.1 to demonstrate this statement.

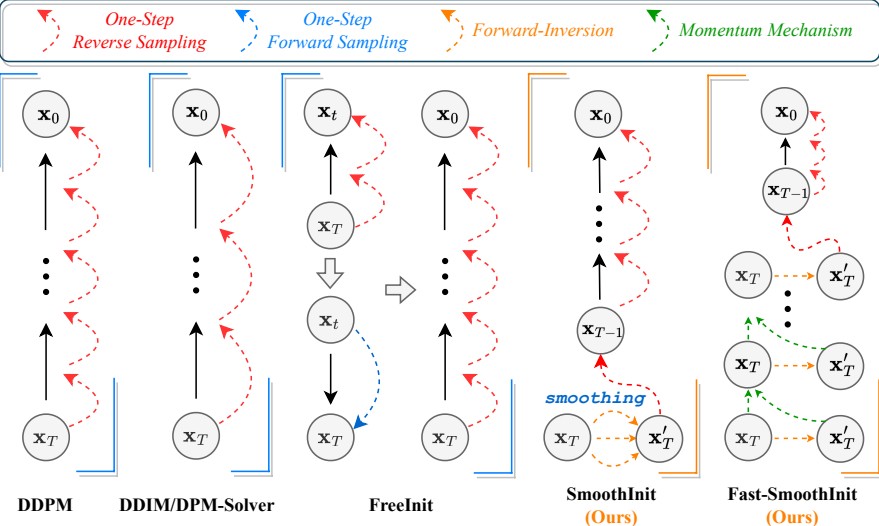

Figure 1: Illustration of SMOOTHINIT and FAST-SMOOTHINIT. Compared with other sampling algorithms, our proposed SMOOTHINIT and FAST-SMOOTHINIT only require optimizing the initial noise, which significantly improves the performance of VDMs.

initial noise in high-dimensional space where the quality of images or videos synthesized from this noise is significantly higher than that of other counterparts. To this end, this work aims to design a novel algorithm to identify the optimal initial noise in the more challenging T2V synthesis task, approaching the problem from the *smoothing* perspective (Cohen et al., 2019).

Identifying *smoothing* as the primary tool for optimizing the initial noise is crucial, as VDMs are built on stochastic processes (Song et al., 2023b), and the randomness inherent in *smoothing* (Cohen et al., 2019) can be effectively combined with stochastic differential equations (SDEs). Thus, it is natural for us to provide a theoretical chain of analysis, combined with empirical visualizations, to elucidate how *smoothing* enhances the quality of synthesized videos. Leveraging the blessing of *smoothing*, we propose a rudimentary algorithm, Smooth⚡nit , along with a more advanced version, Fast-Smooth⚡nit . Specifically, SMOOTHINIT first adds a perturbation to the initial Gaussian noise, then injects semantic information related to the perturbation through the combined action of DDIM (Song et al., 2023a) and DDIM-Inversion (Mokady et al., 2023), and finally computes the expectation of the noise with semantics, conditioned on the perturbation, as the optimized noise. The Corollary 3.4 in this paper proves that SMOOTHINIT is analogous to introducing an additional third-order term on top of gradient descent, leading to more robust convergence. Further, FAST-SMOOTHINIT introduces a momentum mechanism on top of SMOOTHINIT to dynamically update the initial noise before injecting semantic information. This approach not only accelerates convergence but also reduces truncation error with theoretical guarantees (*i.e.*, Theorem 3.5).

We conduct extensive experiments in Sec. 4 on various VDMs, including ANIMATEDIFF (Guo et al., 2023), MODELSCOPE-T2V (Wang et al., 2023), and LATTE (Ma et al., 2024), where the experiments show a consistent and substantial improvement over the widely used baselines (*e.g.*, DDIM) under several popular benchmarks, including UCF-101/MSR-VTT-related FVD (Soomro, 2012), Chronomagic-Bench-150 (Yuan et al., 2024), Chronomagic-Bench-1652 (Yuan et al., 2024), and T2V-Compbench (Sun et al., 2024). Moreover, FAST-SMOOTHINIT outperforms FREEINIT (Wu et al., 2023) and UNICTRL (Chen et al., 2024b) in almost all metrics under the premise of treating the backbone (Ho et al., 2020; Peebles & Xie, 2023) as a black box and only modifying the initial noise, effectively substantiating that SMOOTHNESS is compatible with VDMs.

## 2 BACKGROUND

We present preliminaries on VDMs, DDIM & DDIM-Inversion, the classifier-free guidance and *smoothing* in this section and discuss other related work in detail in Appendix D.

**Video Diffusion models.** VDMs (Blattmann et al., 2023) contain a forward and a reverse process. Denote $\mathbf{x}_0$ a $D$-dimensional random variable from a video data distribution $q_0(\mathbf{x}_0)$. The (discrete)

forward process progressively corrupts "clean video" by interpolating Gaussian noise $\mathbf{x}_T \sim \mathcal{N}(0, \mathbf{I})$ with $\mathbf{x}_0$: $\mathbf{x}_t = \alpha_t \mathbf{x}_0 + \sigma_t \mathbf{x}_T$, where $t \in \{0, 1, \cdots, T\}$ and $\alpha_t$ as well as $\sigma_t$ are pre-defined noise schedule. For the convenience of the derivation in this paper, we define $t \in (0, 1]$ for the (continuous) diffusion process, and $\mathbf{x}_T$ is changed to $\mathbf{x}_1$. Then the (continuous) forward process can be defined as the following stochastic differential equation (SDE):

$$d\mathbf{x}_t = f(t)\mathbf{x}_t dt + g(t)\overline{\boldsymbol{\omega}}_t, \tag{1}$$

where $f(t) = \frac{d \log \alpha_t}{2dt}$ and $g^2(t) = -\frac{d\alpha_t}{dt} - \frac{d \log \alpha_t}{dt}(1 - \alpha_t)$ in the "continuous DDPM" (*i.e.*, VP-SDE (Song et al., 2023b)). $\overline{\boldsymbol{\omega}}_t$ is a standard Wiener process. Every forward process has an equivalent reverse process:

$$d\mathbf{x}_t = \left[ f(t)\mathbf{x}_t - \left[\frac{1 + \lambda^2}{2}\right] g^2(t) \nabla_{\mathbf{x}} \log q_t(\mathbf{x}) \right] dt + \lambda g(t)\overline{\boldsymbol{\omega}}_t, \tag{2}$$

where $\nabla_{\mathbf{x}} \log q_t(\mathbf{x})$ denotes the score function $\nabla_{\mathbf{x}_t} \log p(\mathbf{x}_t)$, and $\overline{\boldsymbol{\omega}}_t$ is a standard Wiener process in backward time. The parameter $\lambda$ is a balancing factor that controls whether the reverse process converges to a SDE (*i.e.*, $1 \geq \lambda > 0$) or an ODE (*i.e.*, $\lambda = 0$). Since $\nabla_{\mathbf{x}} \log q_t(\mathbf{x})$ is unknown during the reverse process, it needs to be replaced by a linear transformation $\frac{\boldsymbol{\epsilon}_\theta(\mathbf{x}_t, t)}{-\sigma_t}$ with the noise estimation model $\boldsymbol{\epsilon}_\theta$. Compared with image data $\mathbf{x}_0 \in \mathbb{R}^{B \times C \times H \times W}$, video data $\mathbf{x}_0 \in \mathbb{R}^{B \times C \times T \times H \times W}$ includes an additional dimension T to represent frames, where B, C, H and W denote the batch size, the number of channels, the height and the width, respectively. This leads to the general treatment of VDMs by using 3D convolutions (*e.g.*, MODELSCOPE-T2V (Wang et al., 2023)), increasing the number of tokens (*e.g.*, LATTE (Ma et al., 2024)), or merging the T dimension into the batch size (*e.g.*, ANIMATEDIFF (Guo et al., 2023)).

**DDIM & DDIM-Inversion.** DDIM (Song et al., 2023a) is an efficient ODE-based sampler that progressively refines the initial Gaussian noise $\mathbf{x}_T$, eventually producing a "clean" video $\mathbf{x}_0$ that follows the data distribution $p_0(\mathbf{x}_0)$. A key feature of DDIM is its *deterministic sampling*, which ensures that the synthesized video is uniquely determined by the initial noise $\mathbf{x}_T$ and the text prompt $\mathbf{c}$. DDIM produces the synthesized data as

$$\mathbf{x}_t = \mathbf{DDIM}(\mathbf{x}_s) = \alpha_t \left( \frac{\mathbf{x}_s - \sigma_s \boldsymbol{\epsilon}_\theta(\mathbf{x}_s, s)}{\alpha_s} \right) + \sigma_t \boldsymbol{\epsilon}_\theta(\mathbf{x}_s, s), \tag{3}$$

where $s$ and $t$ represent timesteps, where $t \leq s$. Using DDIM instead of applying Eq. 1 to add noise under the constraint $t \geq s$ is referred to as DDIM-Inversion. In this paper, we define the operator as $\mathbf{x}_t = \mathbf{DDIM\text{-}Inversion}(\mathbf{x}_s)$.

**Classifier-free Guidance.** Classifier-free guidance (CFG) (Ho & Salimans, 2021) has become a fundamental tool for improving the quality of synthesized data in modern text-guided generation using diffusion models. This technique enhances text conditional guidance through linear transformations of the unconditional score function $\nabla_{\mathbf{x}} \log q_t(\mathbf{x}|\varnothing)$ and the conditional score function $\nabla_{\mathbf{x}} \log q_t(\mathbf{x}|\mathbf{c})$, $\mathbf{c}$ and $\omega$ stand for the text prompt and the CFG scale, respectively:

$$d\mathbf{x}_t = \left[ f(t)\mathbf{x}_t - \left[\frac{1 + \lambda^2}{2}\right] g^2(t) \left[ (\omega + 1) \nabla_{\mathbf{x}} \log q_t(\mathbf{x}|\mathbf{c}) - \omega \nabla_{\mathbf{x}} \log q_t(\mathbf{x}|\varnothing) \right] \right] dt + \lambda g(t)\overline{\boldsymbol{\omega}}_t, \tag{4}$$

**Smoothing.** The essence of *smoothing* is to enhance the stability of the algorithm, with applications such as adversarial attacks and sharpness-aware minimization (SAM). Specifically, consider a classifier $f : [0, 1]^D \to K$ that takes an input $\mathbf{x}_0$ and predicts target class probability over $K$ different classes. Random *smoothing* (Cohen et al., 2019; Yang et al., 2020; Chen et al., 2024a), defined as $g(\mathbf{x}_0)_y = P\left( \arg\max_{\hat{y} \in \{0,1,\cdots,K\}} f(\mathbf{x}_0 + \sigma_{\text{sm}} \cdot \epsilon)_{\hat{y}=y} \right)$ is a theoretical tool to establish a lower bound on robustness against adversarial examples, where $\epsilon \sim \mathcal{N}(0, \mathbf{I})$ is a Gaussian noise and $\sigma_{\text{sm}}$ is the noise level. Then, the lower bound $\underline{p_A}$ and the upper bound $\overline{p_B}$ of $g(\mathbf{x}_0)_y$ are estimated using the Clopper-Pearson lemma. Subsequently, random *smoothing* guarantees that $\mathbf{x}_0$ remains categorically consistent within the certified robust radius $\frac{\sigma_{\text{sm}}\left(\Phi^{-1}(\underline{p_A}) - \Phi^{-1}(\overline{p_B})\right)}{2}$, where $\Phi^{-1}$ refers to the inverse function of the standard Gaussian CDF. In addition, SAM (Chen et al., 2022; Foret et al., 2020; Du et al., 2022) addresses the minimax problem by using the solution of the dual norm, which ensures a smoother loss landscape and ultimately enhances the model's generalization ability.

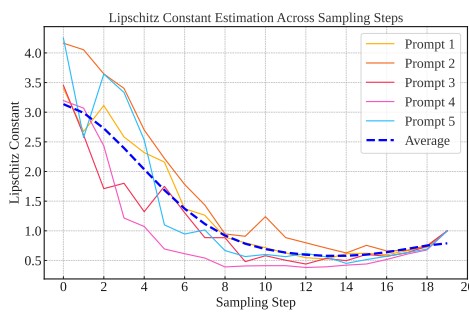
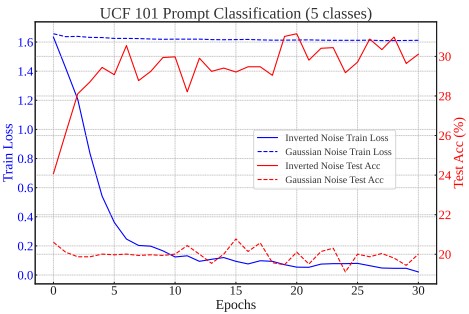

Figure 2: The visualization of $\mathcal{S}$ (*i.e.*, Lipschitz constant) across different timesteps.

Figure 3: Accuracy curves comparing Gaussian noise and optimized noise for classification.

## 3 METHOD

In this section, we first explain why the initial noise is critical, then introduce the base operator used to inject semantic information into the initial Gaussian noise, and finally present both SmoothInit and Fast-SmoothInit, along with their theoretical and empirical justifications.

### 3.1 WHY INITIAL NOISE IS CRUCIAL?

A slight perturbation in the initial noise leads to drastic changes in the synthesized data. We support this claim and demonstrate the importance of initial noise in ensuring the sampling quality of VDMs through Theorem 3.1 and the empirical visualization in Fig. 2.

**Theorem 3.1.** *(the proof in Appendix C.2) Suppose two latents* $\mathbf{x}_H$ *and* $\mathbf{x}'_H$, *the synthesized data* $\mathbf{x}_0$ *and* $\mathbf{x}'_0$ *($H \in \{k, \cdots, T\}$) obtained from* $\mathbf{x}_H$ *and* $\mathbf{x}'_H$ *using DDIM (Song et al., 2023a) sampling with the sampling interval is* $k$ *satisfy* $\|\mathbf{x}_0 - \mathbf{x}'_0\| = \|(\mathbf{x}_H - \mathbf{x}'_H)\mathcal{S}\|$, *where the smoothness factor* $\mathcal{S} = \left[\frac{\alpha_0}{\alpha_H} + \mathbf{Z}(0)\mathbb{E}_{m \sim \textbf{Unif}(1,\cdots,H/k)}\left[\frac{\alpha_0 \alpha_m}{\alpha_{m-k}\alpha_H}\dot{\epsilon}_\theta(\mathbf{x}_m, m)\right] + \mathcal{O}(\mathbf{Z}(0)^2)\right]$ *and* $\mathbf{Z}(t) = \sqrt{1 - \alpha^2_{T-(t+1)k}} - \frac{\sqrt{1-\alpha^2_{T-tk}}\alpha_{T-(t+1)k}}{\alpha_{T-tk}}$ *and* $\|\cdot\|$ *is the Euclidean norm or the Frobenius norm.*

As Appendix C.2 proves, the factor $\mathbf{Z}(t)$ monotonically decreases as $t$ goes from $T$ to $0$. Thus, $\mathbf{Z}(t)$ obtains its maximum value at $t = T$. Considering a very small time interval in which $\alpha_t$ remains constant (*i.e.*, $\alpha_T = \alpha_{T-k}$), the variable $\mathbf{Z}(0)$ is given as $0$. In practice, for models such as STABLE DIFFUSION (SD) V1.5, SD XL, MODELSCOPE-T2V, and ANIMATEDIFF, $\mathbf{Z}(0)$ can be computed as $0.126$. In these cases where $\mathbf{Z}(0)^n \ll 1$ ($n \geq 2$), we can ignore the high-order term $\mathcal{O}(\mathbf{Z}(0)^2)$ in $\mathcal{S}$. Since $\alpha_t \in [0, 1]$ decreases monotonically as $t$ increases, the zero-order term must be significantly larger than $1$, while the first-order term is unknown because $\epsilon_\theta(\cdot, \cdot)$ is a black-box function. We can conclude that the smoothing factor $\mathcal{S}$ decreases as $H$ goes from $T$ to $k$.

To support this theoretical analysis, we present $\mathcal{S}$ estimates across different timesteps on ANIMATEDIFF in Fig. 2, visualizing the dynamic curves for five different prompts along with their average. Each point on the curves represents the effect of the perturbation at the current timestep on the final synthesized video. It is obvious that the initial noise (*i.e.*, $H = T$) has the greatest impact on the sampling output, and this effect decays dramatically as $H$ decrease from $T$ to $k$. Accordingly, our work focuses on optimizing the initial noise to enhance the generative ability of VDMs, leading to strong performance from both our proposed methods SMOOTHINIT and FAST-SMOOTHINIT.

### 3.2 HOW TO OPTIMIZE INITIAL NOISE? FUTURE INFORMATION IS A GOOD REMEDY

How can a desirable perturbation be injected into the initial noise to actively enhance the fidelity of the synthesized data during the reverse process? We address this by selecting a simple yet effective operator that attaches semantic information, which we refer to as FORWARD-INVERSION. This operator can be described as $\mathbf{x}'_T = f_s(\mathbf{x}_T) = \textbf{DDIM-Inversion}(\textbf{DDIM}(\mathbf{x}_T))$, where the sampling path is $\mathbf{x}_T \to \mathbf{x}_{T-\Delta T} \to \mathbf{x}_T$ $s.t., \Delta T > 0$. The key idea is that $f_s(\cdot)$ injects semantic information by leveraging the inconsistency in classifier-free guidance (CFG) between $\textbf{DDIM}(\cdot)$

Table 1: Comparison on UCF-101. Note that UNICTRL (Chen et al., 2024b) does not provide an implementation of ANIMATEDIFF (SD V1.5, MOTION ADAPTER V3) and is tightly coupled to the attention module of the noise estimation model, so it uses the default ANIMATEDIFF (SD V1.5, MOTION ADAPTER V1).

| METHOD | FVD ($\downarrow$) (UCF-101, STYLEGAN) | FVD ($\downarrow$) (UCF-101, VIDEOGPT) | TIME SPENT ($\downarrow$) (S/PER VIDEO) | NO NEED TO ACCESS NOISE ESTIMATION MODEL? | NO NEED TO ACCESS NOISE SAMPLING PROCESS? |
|---|---|---|---|---|---|
| ORIGIN | 815.08 | 819.93 | 21.82 | ✓ | ✓ |
| FORWARD-INVERSION | 797.87 | 801.43 | 22.65 | ✓ | ✓ |
| FREEINIT | 805.33 | 807.04 | 44.66 | ✓ | ✗ |
| UNICTRL | 1757.20 | 1671.43 | 49.10 | ✗ | ✗ |
| SMOOTHINIT$^{30}$ (Ours) | 802.48 | 805.64 | 46.72 | ✓ | ✓ |
| FAST-SMOOTHINIT$^{10}$ (Ours) | 795.83 | 799.27 | 30.12 | ✓ | ✓ |
| FAST-SMOOTHINIT$^{30}$ (Ours) | 717.86 | 721.47 | 46.72 | ✓ | ✓ |

Table 2: Quantitative comparison with popular T2V methods on Chronomagic-Bench-150 (Yuan et al., 2024).

| MODEL | METHOD | $\omega_1 : \omega_2$ | UMT-FVD ($\downarrow$) | UMTSCORE ($\uparrow$) | GPT4O-MTSCORE ($\uparrow$) | MEAN RANK ($\downarrow$) |
|---|---|---|---|---|---|---|
| ANIMATEDIFF (SD V1.5, MOTION ADAPTER V3) | ORIGIN | N/A | 275.18 | 2.82 | 2.83 | 4.00 |
| | FORWARD-INVERSION | 7.5:1 | 267.83 | 2.96 | 2.86 | 4.00 |
| | FREEINIT | N/A | 268.31 | 2.82 | 2.59 | 5.67 |
| | SMOOINIT$^{30}$ (Ours) | 7.5:1 | 259.85 | 3.08 | 2.93 | 2.00 |
| | FAST-SMOOINIT$^{10}$ (Ours) | 7.5:1 | 253.96 | 3.03 | 3.23 | 2.00 |
| | FAST-SMOOINIT$^{30}$ (Ours) | 7.5:1 | 248.61 | 3.04 | 3.00 | 1.67 |
| MODELSCOPE-T2V | ORIGIN | N/A | 241.61 | 2.66 | 2.96 | 5.33 |
| | FORWARD-INVERSION | 7.5:1 | 234.92 | 2.93 | 3.02 | 4.33 |
| | FREEINIT | N/A | 220.96 | 3.01 | 3.09 | 3.00 |
| | SMOOINIT$^{30}$ (Ours) | 7.5:1 | 233.50 | 2.73 | 2.94 | 5.00 |
| | FAST-SMOOINIT$^{10}$ (Ours) | 7.5:1 | 219.72 | 3.06 | 3.19 | 1.00 |
| | FAST-SMOOINIT$^{30}$ (Ours) | 7.5:1 | 219.73 | 3.02 | 3.13 | 2.00 |
| ANIMATEDIFF (SD-XL, BETA) | ORIGIN | N/A | 264.95 | 2.54 | 3.19 | 4.00 |
| | FORWARD-INVERSION | 7.5:1 | 268.38 | 2.45 | 3.20 | 5.00 |
| | FREEINIT | N/A | 256.87 | 2.69 | 3.06 | 3.33 |
| | SMOOINIT$^{30}$ (Ours) | 7.5:7.5 | 284.22 | 2.51 | 3.14 | 6.33 |
| | SMOOINIT$^{30}$ (Ours) | 7.5:1 | 257.76 | 2.69 | 3.19 | 2.67 |
| | FAST-SMOOINIT$^{10}$ (Ours) | 7.5:1 | 270.76 | 2.57 | 3.32 | 3.33 |
| | FAST-SMOOINIT$^{30}$ (Ours) | 7.5:1 | 255.94 | 2.52 | 3.28 | 2.67 |

Table 3: Quantitative comparison with popular T2V methods on Chronomagic-Bench-1649 (Yuan et al., 2024).

| MODEL | METHOD | $\omega_1 : \omega_2$ | UMT-FVD ($\downarrow$) | UMTSCORE ($\uparrow$) | CHSCORE ($\uparrow$) | MEAN RANK ($\downarrow$) |
|---|---|---|---|---|---|---|
| ANIMATEDIFF (SD V1.5, MOTION ADAPTER V3) | ORIGIN | N/A | 219.29 | 3.08 | 11.25 | 4.67 |
| | FORWARD-INVERSION | 7.5:1 | 222.55 | 3.08 | 11.52 | 4.33 |
| | SMOOINIT$^{30}$ (Ours) | 7.5:1 | 219.49 | 3.10 | 11.51 | 3.67 |
| | FAST-SMOOINIT$^{10}$ (Ours) | 7.5:1 | 215.73 | 3.12 | 12.40 | 1.67 |
| | FAST-SMOOINIT$^{30}$ (Ours) | 7.5:1 | 211.45 | 3.11 | 12.48 | 1.33 |

Table 4: Quantitative comparison with popular T2V methods on T2V-Compbench (Sun et al., 2024). CA refers to the CONSIST-ATTR metric and all $\omega_1 : \omega_2$ is set as 7.5 : 1.

| METHOD | ANIMATEDIFF (SD V1.5, MOTION ADAPTER V3) | | | | | MODELSCOPE-T2V | | | |
|---|---|---|---|---|---|---|---|---|---|
| | CA ($\uparrow$) | ACTION ($\uparrow$) | INTERACTION ($\uparrow$) | NUMERACY ($\uparrow$) | AVG ($\uparrow$) | CA ($\uparrow$) | ACTION ($\uparrow$) | NUMERACY ($\uparrow$) | AVG ($\uparrow$) |
| ORIGIN | .6558 | .4159 | .7700 | .2725 | .5286 | .5525 | .4525 | .1891 | .3980 |
| FORWARD-INVERSION | .6645 | .4719 | .7600 | .2647 | .5403 | .5500 | .4699 | .1828 | .4009 |
| SMOOTHINIT$^{30}$ (Ours) | .6350 | .4679 | .7425 | .2659 | .5278 | .5225 | .4759 | .2084 | .4022 |
| FAST-SMOOTHINIT$^{10}$ (Ours) | .6938 | .4539 | .7975 | .2785 | .5559 | .5937 | .4579 | .1897 | .4138 |
| FAST-SMOOTHINIT$^{30}$ (Ours) | .6758 | .4505 | .7800 | .3209 | .5568 | .5813 | .4530 | .2250 | .4198 |

and **DDIM-Inversion**($\cdot$). We denote the CFG sclae in **DDIM**($\cdot$) and **DDIM-Inversion**($\cdot$) as $\omega_1$ and $\omega_2$, respectively. By setting $\omega_1 > \omega_2$, we can naturally stabilize the process and reliably achieve semantic information attachment. To confirm that the optimized noise $x'_T$ indeed contains semantic information, we select the first five prompts (Ge et al., 2023) of UCF-101 in alphabetical order, generate 100 videos with $\omega_1 : \omega_2 = 7.5 : 1$ for each, and split the dataset into a 7:3 ratio for training and testing in classification experiments. The empirical results in Fig. 3 illustrate that $x'_T$ (*i.e.*, "Inverted Noise" in Fig. 3) contains semantic information compared to the initial Gaussian noise $x_T$, thereby increasing the probability of correctly classifying the UCF-101 prompts.

The method of injecting semantic information is not limited to FORWARD-INVERSION. An alternative approach is to directly obtain $x_T + (\omega + 1)\nabla_{\mathbf{x}} \log q_1(\mathbf{x}|\mathbf{c}) - \omega \nabla_{\mathbf{x}} \log q_1(\mathbf{x}|\varnothing)$ from $x_T$ and treat it as the new initial noise. However, this form of gain is neither intuitive nor significant in our exploratory experiments on the image diffusion model (*e.g.*, SD XL), which we attribute to the absence of any introduction of "future" knowledge. By employing a Taylor expansion of the minimum truncation error at $t = 1 - \frac{\eta}{2}$, we provide evidence for this assertion in Theorem 3.2:

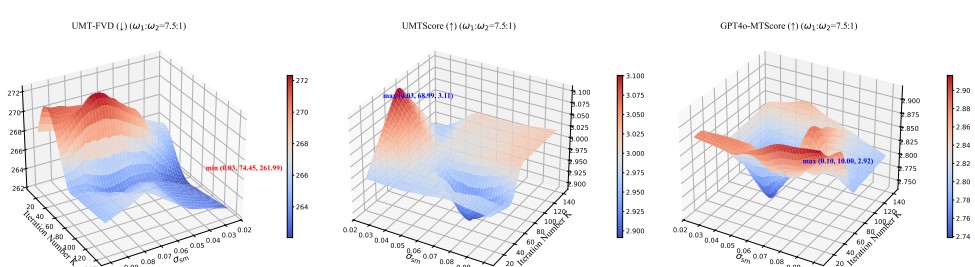

Figure 4: Ablation Studies on hyperparameter the noise level $\sigma_{\text{sm}}$ and the iteration number $K$. Both UMT-FVD and UMTSCORE obtain optimal solutions when $\sigma_{\text{sm}}$ (*i.e.*, 0.03 and 0.03) is small and $K$ (*i.e.*, 74.45 and 68.99) is large. More experiments about $\omega_1 : \omega_2 = 7.5 : 7.5$ can be found in Appendix E.7.

**Theorem 3.2.** *(the proof in Appendix C.4) Given the initial noise* $\mathbf{x}_T \sim \mathcal{N}(0, \mathbf{I})$ *and the operators* **DDIM-Inversion**$(\cdot)$ *and* **DDIM**$(\cdot)$, *and let the* FORWARD-INVERSION *operator as* $f_s(\cdot) =$ **DDIM-Inversion**(**DDIM**$(\cdot)$), *we can obtain that*

$$\mathbf{x}'_T = f_s(\mathbf{x}_T) \implies d\mathbf{x} = -\frac{g^2(1 - \frac{\eta}{2})}{2}\Big[(\omega_1 - \omega_2)\nabla_{\mathbf{x}} \log q_{1 - \frac{\eta}{2}}(\mathbf{c}|\mathbf{x}) + \mathcal{O}(\frac{\eta}{2})\Big] dt, \quad (5)$$

*where* $\mathbf{c}$ *and* $\eta$ *refer to the text prompt and DDIM's single sampling step, respectively.*

We discover that $f_s(\cdot)$ at $t = 1$ (*w.r.t.*, in the continuous scenario) stabilizes and successfully introduces semantic information about the future time point $t = 1 - \frac{\eta}{2}$. It follows that we utilize FORWARD-INVERSION as the base operator in smoothing initialization.

### 3.3 BLESSING OF SMOOTH INITIALIZATION

Relying solely on *Forward-Inversion* to inject "future" semantic information in our experiments (*w.r.t.*, Sec. 4) is insufficient. Therefore, we propose SMOOTHINIT, which we outline in Fig. 1, and it can be formulated as follows:

$$g(\mathbf{x}_T) = \underbrace{\mathbb{E}_{\epsilon \sim \mathcal{N}(0, \sigma_{\text{sm}}^2 \mathbf{I})}[f_s(\mathbf{x}_T + \epsilon)]}_{\text{in theory}} \implies \tilde{g}(\mathbf{x}_T) = \underbrace{\frac{1}{K}\Sigma_{i=1, \epsilon \sim \mathcal{N}(0, \mathbf{I})}^{K}[f_s(\mathbf{x}_T + \sigma_{\text{sm}}\epsilon)]}_{\text{in practice}}, \quad (6)$$

where $\sigma_{\text{sm}}$ and $K$ denote the noise level and the iteration number, respectively. The expectation form is used for theoretical derivation, whereas the summation form is applied for practical reverse sampling. The essence of SMOOTHINIT is to enhance the stability of FORWARD-INVERSION through sufficiently small perturbations, akin to random *smoothing* for achieving certified robustness. We leverage Theorem 3.3 to prove that when the Euclidean distance between $\mathbf{x}$ and $\mathbf{y}$ in high-dimensional space is below a threshold $R$, $g(\mathbf{x})$ and $g(\mathbf{y})$ tend to converge to the same $\epsilon^*$.

**Theorem 3.3.** *(a special case of Cohen et al. (2019, Theorem 1) and proof in Appendix C.5) Given the definitions of* FORWARD-INVERSION *operator* $f_s(\cdot)$ *and* SMOOTHINIT *operator* $g(\cdot)$, $\exists \gamma$ *and* $\epsilon^*$ *satisfy:* $\mathbb{P}(\mathbb{1}[\|f(\mathbf{x} + \epsilon) - \epsilon^*\| \leq \gamma] = 1) \geq \underline{p_1} \geq \overline{p_0} \geq \mathbb{P}(\mathbb{1}[\|f(\mathbf{x} + \epsilon) - \epsilon^*\| \leq \gamma] = 0)$, *where* $\underline{p_1} \in [0, 1]$ *and* $\overline{p_0} \in [0, 1]$. $\mathbb{1}[\cdot]$ *denotes the indicator function. Then* $\mathbb{1}[\|[g(\mathbf{x} + \delta) - \epsilon^*\| \leq \gamma] = 1$ *for all* $\|\delta\| < R$, *where* $R = \frac{\sigma_{\text{sm}}}{2}(\Phi^{-1}(\underline{p_1}) - \Phi^{-1}(\overline{p_0}))$.

Although Theorem 3.3 does not establish that $\epsilon^*$ is indeed preferable for $\mathbf{x}$, it clearly shows that $\mathbf{Z} = g(\mathbf{x})$ is the unique solution to $\mathbb{E}_{(\mathbf{x}, \epsilon) \sim \mathcal{N}(0, \mathbf{I})} \min_{\mathbf{Z}}[\|[f_s(\mathbf{x} + \sigma_{\text{sm}}\epsilon) - \mathbf{Z}\|]$. This also ensures that SMOOTHINIT adheres to the consistency principle, where any two initial Gaussian noises within a distance of $\frac{\sigma_{\text{sm}}}{2}(\Phi^{-1}(\underline{p_1}) - \Phi^{-1}(\overline{p_0}))$ tend to result in consistent optimized noise.

Leveraging the properties of SDEs and *smoothing*, we can gain a deeper understanding of SMOOTHINIT through the optimization theory (Pierre, 1986). To be specific, by eliminating the first-order terms of $g(\cdot)$ using Taylor expansion and utilizing the fact that $\frac{\partial \log q_t(\mathbf{x})}{\partial \mathbf{x}_0}$ can be derived via a linear transformation of $\nabla_{\mathbf{x}} \log q_t(\mathbf{x})$, we arrive at Corollary 3.4:

Table 5: Ablation studies between uniform noise and Gaussian noise for initialization. Additional comparisons between truncated Gaussian noise and standard Gaussian noise are provided in Appendix E.4.

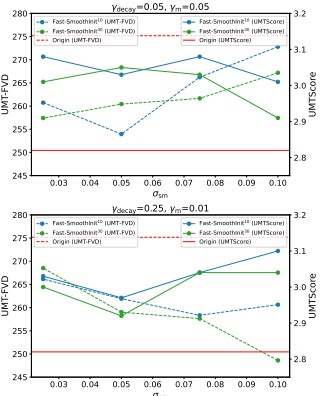

Figure 5: Ablation studies of $\sigma_{\rm sm}$ and $\gamma_{\rm m}$ within FAST-SMOOTHINIT. Due to space constraint, more experiments are provided in Appendix E.5.

| ITERATION NUMBER $K$ | NOISE TYPE | $\sigma_{\rm sm}$ | UMT-FVD (↓) | | UMTSCORE (↑) | |
|---|---|---|---|---|---|---|
| $(\omega_1 : \omega_2)$ | N/A | N/A | (7.5:7.5) | (7.5:1) | (7.5:7.5) | (7.5:1) |
| | Gaussian | 0.100 | 270.91 | 272.53 | 2.93 | 2.96 |
| | Gaussian | 0.075 | 276.76 | 267.45 | 2.93 | 3.07 |
| | Gaussian | 0.050 | 269.28 | 271.40 | 2.91 | 2.98 |
| 30 | Gaussian | 0.025 | 267.29 | 266.55 | 2.94 | 3.02 |
| | Uniform | 0.100 | 271.92 | 268.70 | 2.98 | 2.95 |
| | Uniform | 0.075 | 266.31 | 269.73 | 2.93 | 2.92 |
| | Uniform | 0.050 | 267.92 | 273.54 | 2.87 | 2.94 |
| | Uniform | 0.025 | 264.62 | 259.85 | 2.93 | 3.08 |
| | Gaussian | 0.100 | 262.14 | 264.09 | 2.98 | 3.02 |
| | Gaussian | 0.075 | 274.78 | 267.15 | 2.89 | 3.00 |
| | Gaussian | 0.050 | 265.16 | 266.80 | 3.00 | 2.97 |
| 10 | Gaussian | 0.025 | 268.73 | 265.20 | 2.93 | 2.97 |
| | Uniform | 0.100 | 271.92 | 266.37 | 2.98 | 2.92 |
| | Uniform | 0.075 | 266.30 | 266.14 | 2.93 | 3.03 |
| | Uniform | 0.050 | 267.41 | 261.92 | 2.87 | 3.05 |
| | Uniform | 0.025 | 264.62 | 263.05 | 2.93 | 3.07 |

**Corollary 3.4.** *(the proof in Appendix C.5) Given the definition and conclusion in Theorem 3.2,*
*then* $f_s(\mathbf{x}_T) = \mathbf{x}_T - \frac{\sigma_{T-\frac{1}{2}}}{\alpha_{T-\frac{1}{2}}} \frac{\alpha_{T-1}\sigma_T - \alpha_T\sigma_{T-1}}{\alpha_{T-1}} (\omega_1 - \omega_2) \frac{\partial \left[ -\log q_{1-\frac{\eta}{2}}(\mathbf{x}) \right]}{\partial \mathbf{x}_0}$ *and* $g(\mathbf{x}_T) = f_s(\mathbf{x}_T) -$
$\frac{\sigma_{\rm sm}^2 \sigma_{T-\frac{1}{2}}}{4\alpha_{T-\frac{1}{2}}} \frac{\alpha_{T-1}\sigma_T - \alpha_T\sigma_{T-1}}{\alpha_{T-1}} (\omega_1 - \omega_2) \mathrm{tr} \left( \frac{\partial^3 \left[ -\log q_{1-\frac{\eta}{2}}(\mathbf{x}) \right]}{\partial \mathbf{x}_0^3} \right) + \mathcal{O}\left( \partial^4 \left[ -\log q_{1-\frac{\eta}{2}}(\mathbf{x}) \right] / \partial \mathbf{x}_0^4 \right).$

This Corollary demonstrates that $f_s(\cdot)$ is approximately performing gradient descent to optimize a log likelihood, and $g(\cdot)$ introduces the higher-order term for more accurate gradient descent. To be specific, if $\frac{\sigma_{T-1/2}}{\alpha_{T-1/2}} \frac{\alpha_{T-1}\sigma_T - \alpha_T\sigma_{T-1}}{\alpha_{T-1}} (\omega_1 - \omega_2)$ is considered as the learning rate $\zeta$, then $f_s(\mathbf{x}_T)$ essentially applies gradient descent to decrease the cross-entropy loss $\left[ -\log q_{1-\frac{\eta}{2}}(\mathbf{x}) \right]$ by updating $\mathbf{x}_T$, causing $\mathbf{x}_T$ to move in the direction of $\mathbf{x}_0$. By contrast, $g(\mathbf{x}_T)$ complements $f_s(\mathbf{x}_T)$ with an extra term involving the trace of third-order derivatives, leading to a more precise optimization that helps prevent $\mathbf{x}_T$ from convergent to a suboptimal solution. Additionally, the third-order term corresponds to the Hessian matrix of the *score function* $\nabla_{\mathbf{x}} \log q_t(\mathbf{x})$. Constraining this term with regularization helps reduce the *sharpness* of $\nabla_{\mathbf{x}} \log q_t(\mathbf{x})$ and improve VDMs' generalization ability.

### 3.4 FAST SMOOTH INITIALIZATION WITH THE MOMENTUM MECHANISM

In fact, the complete SMOOTHINIT can be described as $\mathbf{x}'_T = g \circ g \circ \cdots \circ g(\mathbf{x}_T)$, where $\circ$ denotes composition. Compared with vanilla SMOOTHINIT, the complete version requires an excessive number of executions of the *smoothing* operator $g(\cdot)$ to ensure convergence of the gradient descent simulation. This results in significant inference overhead, hindering its practical deployment. Besides, performing $g(\cdot)$ only once would fall into a suboptimal solution. To this end, as shown in Fig. 1, we propose a more advanced version FAST-SMOOTHINIT, which utilizes the momentum mechanism, akin to the momentum technique used in MI-FGSM (Dong et al., 2018), DPM-Solver (Lu et al., 2022c), and Adam (Diederik, 2014). We primarily define two pivotal hyperparameters: the momentum $\gamma_{\rm m}$ and the noise decay rate $\gamma_{\rm decay}$, which control the strength of the initial noise shift and the degree of noise decay at each iteration relative to the previous one, respectively. As shown in Algorithm 1, the key design of FAST-SMOOTHINIT is to achieve more faster optimization by updating the input of $f_s(\cdot)$ per iteration, thus fully utilizing the optimized noise obtained before that iteration.

When the momentum $\gamma_{\rm m}$ in Algorithm 1 is set to 0, it reduces to a special case where the input $\mathbf{x}_T^{\rm tmp}$ of the current iteration becomes the output $f_s(\mathbf{x}_T^{\rm tmp} + \sigma_{\rm sm}(1 - \gamma_{\rm decay})^i \epsilon)$ from the previous iteration. In Theorem 3.5, we conclude that this case can be transformed into an ODE, which differs from traditional diffusion processes such as the Ornstein–Uhlenbeck process.

**Algorithm 1** FAST-SMOOTHINIT ( ⓞ )

---

**Require:** The FORWARD-INVERSION operator $f_s(\cdot)$; the noise level $\sigma_{\text{sm}}$; the iteration number $K$; the initial Gaussian noise $\mathbf{x}_T$; the momentum $\gamma_{\text{m}}$; the noise decay rate $\gamma_{\text{decay}}$.

1: **Initialize:** The previous noise is set as $\mathbf{m}_0 \leftarrow \mathbf{x}_T$; the noise from two steps ago as $\mathbf{m}_1 \leftarrow \mathbf{x}_T$; and the noise from three steps ago as $\mathbf{m}_2 \leftarrow \mathbf{x}_T$.
2: **for** $i = 0$ to $K$ **do**
3:    $\mathbf{x}_T^{\text{tmp}} \leftarrow (1 - \gamma_{\text{m}})\mathbf{m}_0 + (1 - \gamma_{\text{m}})\gamma_{\text{m}}\mathbf{m}_1 + \gamma_{\text{m}}^2\mathbf{m}_2.$    ▶ Obtain the input of $f_s(\cdot)$
4:    $\mathbf{x}_T^{\text{tmp}} \leftarrow f_s(\mathbf{x}_T^{\text{tmp}} + \sigma_{\text{sm}}(1 - \gamma_{\text{decay}})^i \epsilon)$, where $\epsilon \sim \mathcal{N}(0, \mathbf{I})$.    ▶ Inject semantic information
5:    $\mathbf{m}_2 \leftarrow \mathbf{m}_1; \mathbf{m}_1 \leftarrow \mathbf{m}_0; \mathbf{m}_0 \leftarrow \mathbf{x}_T^{\text{tmp}}.$    ▶ Update $\mathbf{m}_2, \mathbf{m}_1$ and $\mathbf{m}_0$
6: **end for**
7: **Return:** The optimized noise $\mathbf{m}_0$.

---

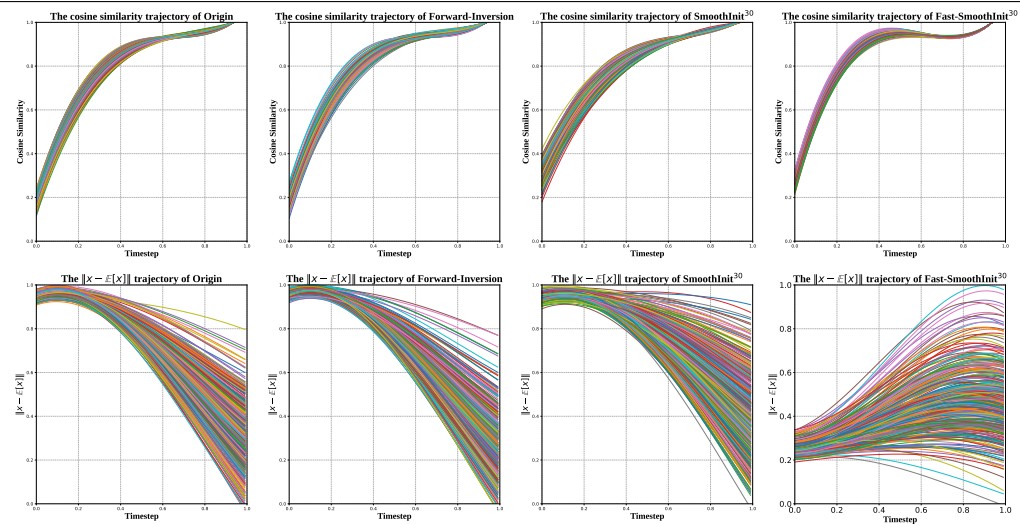

Figure 6: Visualization of 2D sampling trajectories in MODELSCOPE-T2V. The prompt is "Spiderman is surfing". Each subgraph visualizes 500 sampling trajectories. More visualization can be found in Appendix E.2.

**Theorem 3.5.** *(the proof in Appendix C.6) Based on the conclusion of Theorem 3.2,* FAST-SMOOTHINIT *can be reformulated as an ODE:*

$$d\mathbf{x} = \left[ -\frac{\sigma_{T-\frac{1}{2}}}{\Delta t} \frac{\alpha_{T-1}\sigma_T - \alpha_T\sigma_{T-1}}{\alpha_{T-1}}(\omega_1 - \omega_2) + \frac{1}{2\Delta t}\sigma_{\text{sm}}^2(1 - \gamma_{\text{decay}})^{2i} \right] \nabla_{\mathbf{x}} \log q_t(\mathbf{x})dt, \quad (7)$$

where $\Delta t = \frac{1}{K}$. In particular, we can substantiate that FAST-SMOOTHINIT and DPM-Solver share similarities in reducing truncation errors. To be specific, by calculating the analytic solution of Eq. 7 over a single sampling interval and approximating it using a Taylor expansion (Lu et al., 2022c), the process eventually becomes FAST-SMOOTHINIT with a momentum mechanism. Unlike DPM-Solver, which determines the optimal hyperparameters theoretically. Motivated by Zheng et al. (2023), FAST-SMOOTHINIT relies on a heuristic grid search to find the optimal $\gamma_{\text{m}}$ and $\gamma_{\text{decay}}$.

## 4 EXPERIMENT

### 4.1 IMPLEMENTATION DETAILS

To assess the effectiveness of SMOOTHINIT and FAST-SMOOTHINIT in improving fidelity, interframe consistency, alignment between the synthesized video and the text prompt, and diversity, we apply our proposed methods to four publicly available diffusion-based T2V VDMs: ANIMATEDIFF (SD V1.5, MOTION ADAPTER V3) (Guo et al., 2023), ANIMATEDIFF (SD XL, MOTION ADAPTER BETA) (Guo et al., 2023), MODELSCOPE-T2V (Wang et al., 2023), and LATTE (Ma et al., 2024). For traditional video metrics, we follow Wu et al. (2023) and Ge et al. (2023), which assess sampling performance (*i.e.*, Fréchet Video Distance, FVD (Unterthiner et al., 2019)) using prompts from the UCF-101 (Soomro, 2012) and MSR-VTT (Xu et al., 2016) datasets. Additionally, we evaluate several new benchmarks that are popular on the *AIGC community*, including Chronomagic-Bench-150 (Yuan et al., 2024), Chronomagic-Bench-1649 (Yuan et al., 2024),

and T2V-Compbench (Sun et al., 2024). Specifically, Chronomagic-Bench-150 and Chronomagic-Bench-1649 include five metrics: UMT-FVD ($\downarrow$), MTSCORE ($\uparrow$), UMTSCORE ($\uparrow$), CHSCORE ($\uparrow$), and GPT4O-MTSCORE ($\uparrow$). These metrics assess various aspects: visual quality, frame-to-frame variation, alignment between the synthesized video and the text prompt, temporal coherence, as well as both metamorphic amplitude and temporal coherence, respectively. T2V-Compbench is a compositional benchmark that primarily focuses on consistent attribute binding (*i.e.*, CONSIST-ATTR ($\uparrow$)), action binding (*i.e.*, ACTION ($\uparrow$)), object interaction (*i.e.*, INTERACTION ($\uparrow$)), and generative numeracy (*i.e.*, NUMERACY ($\uparrow$)). *We provide a detailed description of the benchmarks and VDMs in Appendix A.1 and Appendix A.2. The limitations can be found in Appendix B.*

In our experiments, we apply three different hyperparameter settings to implement SMOOTH-INIT and FAST-SMOOTHINIT: SMOOTHINIT$^{30}$, FAST-SMOOTHINIT$^{10}$, and FAST-SMOOTHINIT$^{30}$. These settings are derived from ablation studies discussed later, where the superscript $^{[\text{number}]}$ indicates the number of iterations $K$. *For the exact settings, please refer to Appendix A.3.*

### 4.2 QUANTITATIVE COMPARISON

**Traditional Bench.** We compare SMOOTHINIT and FAST-SMOOTHINIT to the standard DDIM process, the FORWARD-INVERSION operator, FREEINIT, and UNICTRL across various VDMs and benchmarks. We evaluate the inference performance of these algorithms on the UCF-101 and MSR-VTT datasets, using Open-Sora-Plan's (Lab & etc., 2024) implementation of FVD (Unterthiner et al., 2019). The results for UCF-101 and MSR-VTT are shown in Table 1 and Fig.7's *Bottom-Right*, respectively. As indicated in Table1, our proposed methods consistently outperform FREEINIT and UNICTRL without requiring access to $\epsilon_\theta(\cdot, \cdot)$ or the standard reverse process, achieving state-of-the-art (SOTA) performance. Notably, FAST-SMOOTHINIT$^{30}$ surpasses standard DDIM sampling by approximately 100 points. Similarly, both SMOOTHINIT$^{30}$ and FAST-SMOOTHINIT$^{30}$ significantly enhance the sampling performance of MODELSCOPE-T2V on MSR-VTT-related FVD.

**Chronomagi-Bench-150 & Chronomagi-Bench-1649.** As shown in Table 2, SMOOTHINIT$^{30}$, FAST-SMOOTHINIT$^{10}$, and FAST-SMOOTHINIT$^{30}$ achieve the best performance across all metrics on Chronomagi-Bench-150, particularly excelling in UMTSCORE (*w.r.t.*, the semantic faithfulness) and GPT4O-SCORE (*w.r.t.*, the metamorphic amplitude and temporal coherence), significantly surpassing the previous SOTA method, FREEINIT. Similarly, as presented in Table 3, FAST-SMOOTHINIT$^{10}$ and FAST-SMOOTHINIT$^{30}$ also secured both the winner and runner-up across all metrics on Chronomagi-Bench-1649.

**T2V-Compbench.** We also evaluate the generative capability of our proposed method on the widely-used compositional T2V benchmark. As shown in Table 4, our methods achieve the strongest performance across three metrics: CONSIST-ATTR, INTERACTION, and NUMERACY. Notably, as illustrated in Fig. 7, SMOOTHINIT and FAST-SMOOTHINIT significantly outperform the standard DDIM process in NUMERACY when applied to the text prompt, "Two cats chase $\cdots$ sunny garden."

### 4.3 ABLATION STUDY

We demonstrate in Fig. 4 that the optimal settings for $\sigma_{\text{sm}}$ and $K$ in SMOOTHINIT vary depending on the metric. A clear pattern emerges: UMT-FVD and UMTSCORE improve with larger $K$ and smaller $\sigma_{\text{sm}}$, while GPT4O-SCORE performs better with smaller $K$ and larger $\sigma_{\text{sm}}$. Considering computational overhead, we adopt $K = 30$ and the corresponding $\sigma_{\text{sm}}$ as the default SMOOTHINIT configuration. Furthermore, we investigate the effect of $\gamma_{\text{m}}$ and $\gamma_{\text{decay}}$ on video quality using grid search. As shown in Fig. 5 and Appendix E.5, FAST-SMOOTHINIT consistently outperforms the vanilla sampling method across nearly all settings. The optimal parameters are $\gamma_{\text{m}} = 0.05, \gamma_{\text{decay}} = 0.05$ for FAST-SMOOTHINIT$^{10}$ and $\gamma_{\text{m}} = 0.01, \gamma_{\text{decay}} = 0.25$ for FAST-SMOOTHINIT$^{30}$. Finally, inspired by Xie et al. (2021), we attempt to use uniform noise from $\mathcal{U}[-\sqrt{3}, \sqrt{3}]$ instead of Gaussian noise to achieve more stable optimization. As shown in Table 5, this approach slightly reduces UMT-FVD. However, it degrades the performance of FAST-SMOOTHINIT, so we apply the technique exclusively to SMOOTHINIT. Due to space constraints, we present and discuss additional ablation experiments, including but not limit to the combination of our methods with DPM-Solver++, the additional computational cost, and experiments on the LATTE (*w.r.t.*, DiT), in Appendix E.

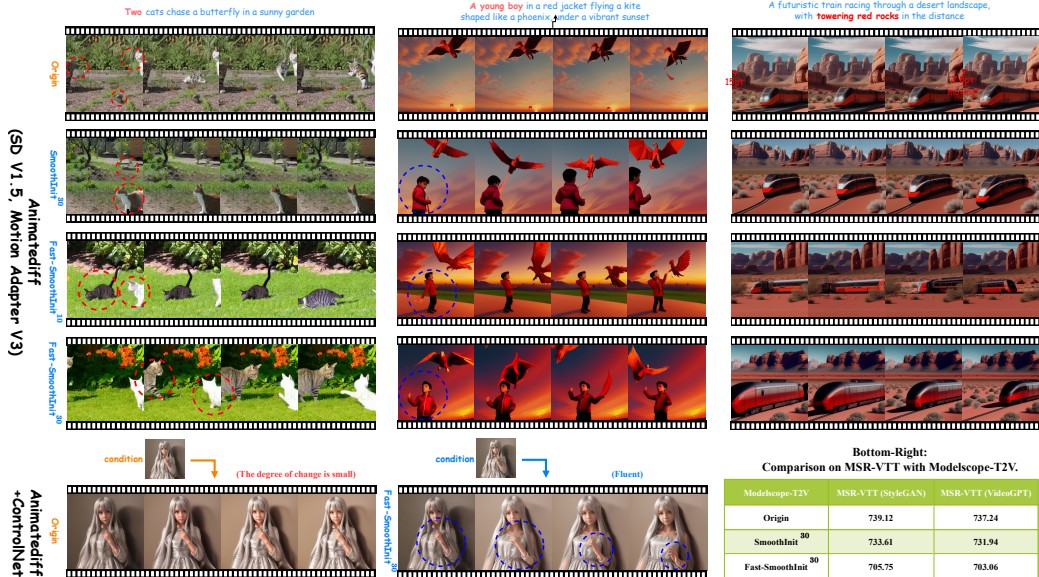

Figure 7: Visualization of SMOOTHINIT, FAST-SMOOTHINIT and the standard DDIM on ANIMATEDIFF. *Bottom-Right:* Comparison on the MSR-VTT (Xu et al., 2016) dataset with MODELSCOPE-T2V.

## 4.4 VISUALIZATION

**Sampling Trajectory.** We visualize the sampling trajectories of different processes initiated by various types of noise, including Origin, FORWARD-INVERSION, SMOOTHINIT[30], and FAST-SMOOTHINIT[30]. The *top* of Fig. 6 shows the cosine similarity curve $\frac{\langle \mathbf{x}_t - \mathbf{x}_T, \mathbf{x}_0 - \mathbf{x}_T \rangle}{\|\mathbf{x}_t - \mathbf{x}_T\| \cdot \|\mathbf{x}_0 - \mathbf{x}_T\|}$ at each time step $t$. At the start of the sampling process (timestep = 0 in Fig. 6), the similarity for FAST-SMOOTHINIT[30] is higher than that of SMOOTHINIT[30] and the other methods, indicating that the sampling trajectory of FAST-SMOOTHINIT[30] is the most straight (Liu et al., 2022). This suggests that the video synthesized by FAST-SMOOTHINIT[30] is closer to the real data distribution (Villani et al., 2009). Moreover, the *bottom* of Fig. 6 presents the curve showing the distance of $\mathbf{x}_t$ from the mean $\mathbb{E}[\mathbf{x}_t]$. We observe that FAST-SMOOTHINIT behaves significantly differently from the other three methods. Specifically, at the endpoint of the sampling process (timestep = 1 in Fig.6), the video generated by FAST-SMOOTHINIT tends to be more uniformly distributed within a sphere, whereas SMOOTHINIT, FORWARD-INVERSION, and Origin tend to be distributed along a spherical shell.

**Synthesized Video.** Fig. 7 presents the synthesized videos from different noise using ANIMATEDIFF and the combination of ANIMATEDIFF with CONTROLNET (Zhang et al., 2023). Across various prompts, both SMOOTHINIT and FAST-SMOOTHINIT significantly enhance the fidelity and semantic faithfulness of the synthesized videos compared to Origin. For example, our proposed methods effectively render the prompt "A young boy ⋯ flying a kite ⋯", whereas pure Gaussian noise with standard DDIM sampling fails to do so. More visualization can be found in Appendix F.

## 5 CONCLUSION

In this paper, we extensively explore initial noise optimization to improve the quality and fidelity of synthesized videos from a *smoothing* perspective. Through empirical investigations and theoretical analysis, we confirm the significant sensitivity of the final video to the initial noise and demonstrate the FORWARD-INVERSION operator's ability to consistently optimize this noise by injecting semantic information. Building on these insights, we propose a basic algorithm, SMOOTHINIT, which improves optimization by accounting for bias introduced by intra-closed ball perturbations, thereby enhancing VDM generation quality. We further design an advanced version, FAST-SMOOTHINIT, which achieves better performance and faster convergence by dynamically updating the input to the FORWARD-INVERSION operator during iterations. We hope that our exploration of *smoothing* in VDMs inspires future research on more efficient noise initialization techniques, contributing to the *AIGC community*.

**Ethics Statement.** We introduce ⚡Smooth⚡nit and Fast-Smooth⚡nit , two approaches aimed at improving the semantic accuracy and visual quality of video generated by Video Diffusion Models. While our methods do not directly involve human participants, we remain dedicated to ensuring their responsible use, prioritizing respect for user autonomy and fostering positive outcomes. In line with data protection standards, we emphasize the privacy and security of any synthesized videos and prompts involved in our work. Recognizing the commercial potential of ⚡Smooth⚡nit and Fast-Smooth⚡nit , we are committed to their ethical deployment, with a focus on maximizing societal benefits and minimizing potential risks.

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

## A    ADDITIONAL INPLEMENTATION DETAILS

### A.1    BENCHMARKS

Here we present the relevant metrics and benchmarks used for comparison in the main paper.

**UCF-101-related FVD.**    The UCF-101 dataset is an action recognition dataset consisting of 101 categories. All videos are sourced from YouTube, with a fixed frame rate of 25 frames per second (FPS) and a resolution of 320×240. Several previous works (Blattmann et al., 2023; Wu et al., 2023; Chen et al., 2024b) validate the generation performance of VDMs on the UCF-101 dataset using FVD (Unterthiner et al., 2019). However, a comprehensive evaluation benchmark for UCF-101 remains unavailable. Consequently, we follow FREEINIT, using the prompts listed in Ge et al. (2023) to synthesize videos and evaluate inference performance with FVD. Specifically, we synthesize 5 videos for each of the 101 prompts provided by Ge et al. (2023), resulting in 505 synthesized videos. We then compute the FVD between these 505 synthesized videos and 505 videos randomly sampled from the UCF-101 dataset (5 per class) by Open-Sora-Plan's[2] built-in FVD evaluation code.

**MSR-VTT-related FVD.**    The MSR-VTT dataset (Xu et al., 2016) is a large-scale dataset for open-domain video captioning, consisting of 10,000 video clips from 20 categories. The standard split includes 6,513 clips for training, 497 clips for validation, and 2,990 clips for testing. We used all 497 validation videos for our measurement. First, we generate a total of 1,491 videos based on the prompts from the 497 validation videos, with each prompt generating three different videos to ensure evaluation stability. We finally use Open-Sora-Plan's[3] built-in FVD evaluation code to assess the results.

**Chronomagic-Bench-150.**    Chronomagic-Bench-150, proposed in (Yuan et al., 2024), is a comprehensive benchmark with a primary focus on the metamorphic evaluation of timelapse T2V synthesis. This benchmark includes four major categories of time-lapse videos: biological, human-created, meteorological, and physical, and extends these into 75 subcategories. Each subcategory contains two hard prompts, resulting in a total of 150 prompts. Chronomagic-Bench-150 comprises four metrics: UMT-FVD ($\downarrow$), MTSCORE ($\uparrow$), UMTSCORE ($\uparrow$), and GPT4O-MTSCORE ($\uparrow$), each used to evaluate different aspects. More specifically, UMT-FVD ($\downarrow$) (Liu et al., 2024b) uses the UMT (Li et al., 2023) feature space to compute FVD and evaluate the visual quality of the synthesized video. MTSCORE ($\uparrow$) measures metamorphic amplitude, reflecting the degree of change across frames. UMTSCORE ($\uparrow$) employs the UMT (Li et al., 2023) feature space to compute CLIPScore (Hessel et al., 2021), assessing the text relevance of the synthesized video. Finally, GPT4O-MTSCORE ($\uparrow$) is a fine-grained metric that uses GPT-4o (Achiam et al., 2023) as an evaluator, aligning with human perception to accurately reflect the metamorphic amplitude and temporal coherence of T2V models. In this paper, we use metrics UMT-FVD ($\downarrow$), UMTSCORE ($\uparrow$), and GPT4O-MTSCORE ($\uparrow$) because we find that MTSCORE ($\uparrow$) exhibits a peculiar phenomenon: performance decreases as the number of sampling steps increases (from 30 to 50), which may introduce ambiguity.

**Chronomagic-Bench-1649.**    Chronomagic-Bench-1649, proposed in (Yuan et al., 2024), is a comprehensive benchmark with a primary focus on the metamorphic evaluation of timelapse T2V synthesis. This benchmark has 75 subcategories like Chronomagic-Bench-150 but has 1649 prompts, which is more comprehensive compared to the lightweight benchmark Chronomagic-Bench-150. Chronomagic-Bench-1649 comprises four metrics: UMT-FVD ($\downarrow$), MTSCORE ($\uparrow$), UMTSCORE ($\uparrow$), and CHSCORE ($\uparrow$), each used to evaluate different aspects. More specifically, UMT-FVD ($\downarrow$) (Liu et al., 2024b) uses the UMT (Li et al., 2023) feature space to compute FVD and evaluate the visual quality of the synthesized video. MTSCORE ($\uparrow$) measures metamorphic amplitude, reflecting the degree of change across frames. UMTSCORE ($\uparrow$) employs the UMT (Li et al., 2023) feature space to compute CLIPScore (Hessel et al., 2021), assessing the text relevance of the synthesized video. Finally, CHSCORE ($\uparrow$) evaluates temporal coherence, ensuring that the generated videos maintain logical progression and continuity. Similar to Chronomagic-Bench-150, we ignore the metric MTScore ($\uparrow$) in our experiments.

---

[2]https://github.com/PKU-YuanGroup/Open-Sora-Plan
[3]https://github.com/PKU-YuanGroup/Open-Sora-Plan

**T2V-Compbench.** TV2-Compbench (Sun et al., 2024) is a benchmark specifically designed for compositional text-to-video (T2V) synthesis. It covers various aspects of compositionality, including consistent attribute binding, dynamic attribute binding, spatial relationships, motion binding, action binding, object interactions, and generative numeracy. TV2-Compbench includes seven distinct metrics, with each metric consisting of 100 prompts. In this paper, we focus on four metrics: CONSIST-ATTR ($\uparrow$), ACTION ($\uparrow$), INTERACTION ($\uparrow$), and NUMERACY ($\uparrow$). Specifically, CONSIST-ATTR ($\uparrow$) evaluates the consistent attribute binding ability of VDMs, assessed by LLAVA-34b (Liu et al., 2024a); ACTION ($\uparrow$) evaluates the action binding ability of VDMs, also assessed by LLAVA-34b (Liu et al., 2024a); INTERACTION ($\uparrow$) evaluates the object interaction ability of VDMs, again assessed by LLAVA-34b (Liu et al., 2024a); and NUMERACY ($\uparrow$) assesses the generative numeracy ability of VDMs, evaluated by GroundingSAM (Ren et al., 2024).

## A.2 VIDEO DIFFUSION MODELS

We describe the VDMs utilized in this work. Specifically, we employ three VDMs with distinct architectures: MODELSCOPE-T2V (Wang et al., 2023), ANIMATEDIFF (Guo et al., 2023), and LATTE (Ma et al., 2024):

**Modelscope-T2V.** MODELSCOPE-T2V incorporates spatio-temporal blocks to ensure consistent frame generation and smooth motion transitions. Its most critical features include the use of 3D convolution, training from scratch. The input video size is $3 \times 16 \times 256 \times 256$, where 3 represents the number of channels, 16 is the number of frames, and $256 \times 256$ refers to the resolution.

**Animatediff.** ANIMATEDIFF does not require training from scratch. It only needs fine-tuning on existing image diffusion models. ANIMATEDIFF's motion adapter is a plug-and-play module that converts most community text-to-image models into animation generators. In this paper, we use two different versions, ANIMATEDIFF (SD V1.5, MOTION ADAPTER V3) and ANIMATEDIFF (SD XL, BETA). The former was fine-tuned from SD V1.5, while the latter was fine-tuned from SD XL. The input video size of ANIMATEDIFF (SD V1.5, MOTION ADAPTER V3) is $3 \times 16 \times 512 \times 512$, where 3 represents the number of channels, 16 is the number of frames, and $512 \times 512$ refers to the resolution. The input video size of ANIMATEDIFF (SD XL, BETA) is $3 \times 16 \times 1024 \times 1024$, where 3 represents the number of channels, 16 is the number of frames, and $1024 \times 1024$ refers to the resolution.

**Latte.** LATTE is a scratch-trained VDM built on the diffusion transformer (DiT). Unlike MODELSCOPE-T2V and ANIMATEDIFF, both the VAE Encoder and VAE Decoder of LATTE are retrained specifically for T2V synthesis. The input video size is $3 \times 16 \times 512 \times 512$, where 3 represents the number of channels, 16 is the number of frames, and $512 \times 512$ refers to the resolution.

## A.3 HYPERPARAMETER SETTINGS

For all VDMs, we use the default sampling steps, CFG scale, and sampler from their respective papers or demos. To be specific, for MODELSCOPE-T2V, the sampling steps, CFG scale (*i.e.*, $\omega_1$), and sampler are set to 50, 7.5, and **DDIM**($\cdot$), respectively. Similarly, for ANIMATEDIFF (SD V1.5, MOTION ADAPTER V3), these parameters are also set to 50, 7.5, and **DDIM**($\cdot$). For ANIMATEDIFF (SD XL, BETA), the sampling steps, CFG scale (*i.e.*, $\omega_1$), and sampler are set to 20, 7.5, and **DDIM**($\cdot$). Lastly, for LATTE, the sampling steps, CFG scale (*i.e.*, $\omega_1$), and sampler are set to 50, 7.5, and **DDIM**($\cdot$).

For SMOOTHINIT, the noise level $\sigma_{\text{sm}}$, the number of iterations $K$, and the noise type are set to 0.025, 30, and uniform noise $\mathcal{U}\left[-\sqrt{3}, \sqrt{3}\right]$, respectively. For FAST-SMOOTHINIT[10], the noise level $\sigma_{\text{sm}}$, the number of iterations $K$, and the noise type are set to 0.05, 10, and Gaussian noise $\mathcal{N}(0, \text{I})$, respectively. Specifically, $\gamma_{\text{m}}$ and $\gamma_{\text{decay}}$ are both set to 0.05. For FAST-SMOOTHINIT[30], the noise level $\sigma_{\text{sm}}$, the number of iterations $K$, and the noise type are set to 0.1, 30, and Gaussian noise $\mathcal{N}(0, \text{I})$, respectively. Specifically, $\gamma_{\text{m}}$ and $\gamma_{\text{decay}}$ are set to 0.01 and 0.25, respectively.

We found that on LATTE, both FAST-SMOOTHINIT[10] and FAST-SMOOTHINIT[30] are prone to negative optimization under the hyperparameter settings mentioned above. Upon analysis, we determined that this is because LATTE is significantly less tolerant of the effective range of the initial

noise $\mathbf{x}_T$. To address this issue, we increased $\gamma_{\mathrm{m}}$ to $0.95$ in both FAST-SMOOTHINIT[10] and FAST-SMOOTHINIT[30], ensuring stability and eventually achieving the desired optimization.

## B    LIMITATION

Although SMOOTHINIT and FAST-SMOOTHINIT significantly enhance the performance of the final synthesized video in VDMs, they generally introduce additional inference overhead. This overhead is less substantial compared to FREEINIT and UNICTRL, but it still presents deployment challenges relative to standard DDIM sampling. Additionally, the conditions under which SMOOTHINIT and FAST-SMOOTHINIT may negatively impact certain metrics remain unexplored. Finally, while our exploratory experiments indicate that FAST-SMOOTHINIT performs well on image synthesis tasks, whether the algorithm can be extended to 3D rendering and graph generation remains an open question.

## C    THEORETICAL ANALYSIS

### C.1    ASSUMPTIONS

Throughout this section, we adopt the regularity assumptions from Lu et al. (2022a, Assumption A.1) and Nie et al. (2024, Assumption D.1). These technical assumptions guarantee the existence of a solution for smooth initialization in diffusion sampling and ensure the validity of integration by parts and the Fokker-Planck equations. For completeness, we list these assumptions in this section.

**Assumption C.1.** *We make two assumptions from Lu et al. (2022a, Assumption A.1) and Nie et al. (2024, Assumption D.1), and we include them here only for completeness:*

1. $\exists C > 0, \forall \boldsymbol{x}, \boldsymbol{y} \in \mathbb{R}^d : \|\nabla_t \log q_t(\mathbf{x}) - \nabla_t \log q_t(\mathbf{y})\|_2 \leq C\|\boldsymbol{x} - \boldsymbol{y}\|_2$.

2. $\forall t \in [0, T], \exists k > 0 : q_t(\boldsymbol{x}) = O(e^{-\|\boldsymbol{x}\|_2^k}), p_t^{SDE}(\boldsymbol{x}) = O(e^{-\|\boldsymbol{x}\|_2^k}), p_t^{ODE}(\boldsymbol{x}) = O(e^{-\|\boldsymbol{x}\|_2^k})$ as $\|\boldsymbol{x}\|_2 \to \infty$.

### C.2    THE IMPORTANCE OF INITIAL NOISE

Here, we provide a theoretical proof of Theorem 3.1 from an error analysis perspective. Additionally, we demonstrate that Theorem 3.1 holds for the remaining models in the stable diffusion family, except for stable diffusion (SD) V3, along with an in-depth corresponding interpretation. Finally, we present the process of transforming the entropy of $p_t(\mathbf{x})$ into the counterpart of $p_0(\mathbf{x})$ using the Fokker-Planck equation.

*Proof.* The widely adopted DDIM sampler can be denoted as

$$\mathbf{x}_t = \mathbf{DDIM}(\mathbf{x}_s) = \alpha_t \left( \frac{\mathbf{x}_s - \sqrt{1 - \alpha_s^2}\boldsymbol{\epsilon}_\theta(\mathbf{x}_s, s)}{\alpha_s} \right) + \sqrt{1 - \alpha_t^2 - \sigma_s^2}\boldsymbol{\epsilon}_\theta(\mathbf{x}_s, s) + \sigma_s \overline{\boldsymbol{\omega}}_s, \quad (8)$$

where $s$ and $t$ represent timesteps with $t \leq s$ and $\overline{\boldsymbol{\omega}}_s$ is a standard Gaussian noise independent of $\mathbf{x}_s$. The term $\sigma_s = \eta\sqrt{(1 - \alpha_t^2)/(1 - \alpha_s^2)}\sqrt{1 - \alpha_s^2/\alpha_t^2}$ controls the form of the differential equation during the backward sampling process, determining whether it is an SDE or an ODE. When $\eta$ is set to 0, DDIM Sampler reduces to the *deterministic sampling* method used by default in this paper.

Suppose two Gaussian noise (vectors) $\mathbf{x}_T$ and $\mathbf{x}_T'$ used for the initialization of reverse sampling, where $\mathbf{x}_T' = \mathbf{x}_T + \Delta\mathbf{x}_T$, $\Delta\mathbf{x}_T$ is introduced to determine the scaling of the error during sampling. Given this, we can further obtain the following derivation:

$$\frac{\mathbf{DDIM}(\mathbf{x}_T') - \mathbf{DDIM}(\mathbf{x}_T)}{\Delta\mathbf{x}_T} = \frac{\alpha_t}{\alpha_T} + \left[ \sqrt{1 - \alpha_t^2} - \frac{\sqrt{1 - \alpha_T^2}\alpha_t}{\alpha_T} \right] \dot{\boldsymbol{\epsilon}}_\theta(\mathbf{x}_T, T). \quad (9)$$

Assume that the DDIM Sampler need to sample $N$ steps in practical application and the interval between two neighbouring steps is $k$, *i.e.*, $Nk = T$. In this case, the relative error at the $T-k$ moment can be expressed as

$$\left\|\frac{\Delta \mathbf{x}_{T-k}}{\Delta \mathbf{x}_T}\right\| = \left\|\frac{\alpha_{T-k}}{\alpha_T} + \left[\sqrt{1-\alpha_{T-k}^2} - \frac{\sqrt{1-\alpha_T^2}\alpha_{T-k}}{\alpha_T}\right]\dot{\boldsymbol{\epsilon}}_\theta(\mathbf{x}_T, T)\right\|. \tag{10}$$

Similarly, we can obtain the iterative equation for the timestep $t \in \{k, 2k, \cdots, T\}$:

$$\left\|\frac{\Delta \mathbf{x}_{t-k}}{\Delta \mathbf{x}_t}\right\| = \left\|\frac{\alpha_{t-k}}{\alpha_t} + \left[\sqrt{1-\alpha_{t-k}^2} - \frac{\sqrt{1-\alpha_t^2}\alpha_{t-k}}{\alpha_t}\right]\dot{\boldsymbol{\epsilon}}_\theta(\mathbf{x}_t, t)\right\|. \tag{11}$$

Through the recursive method we can get the relative error at $\mathbf{x}_0$:

$$\left\|\frac{\Delta \mathbf{x}_0}{\Delta \mathbf{x}_T}\right\| = \left\|\prod_{i=0}^{N-1}\left[\frac{\alpha_{T-(i+1)k}}{\alpha_{T-ik}} + \left(\sqrt{1-\alpha_{T-(i+1)k}^2} - \frac{\sqrt{1-\alpha_{T-ik}^2}\alpha_{T-(i+1)k}}{\alpha_{T-ik}}\right)\dot{\boldsymbol{\epsilon}}_\theta(\mathbf{x}_{T-ik}, T-ik)\right]\right\| \tag{12}$$

As a differentiable function, $\alpha_t$ increases monotonically as $t$ approaches 0. Consequently, $\frac{\alpha_{T-(i+1)k}}{\alpha_{T-ik}}$ is always greater than 1 for $i \in \{0, 1, \cdots, N-1\}$. The first term $\prod_{i=0}^{N-1}\frac{\alpha_{T-(i+1)k}}{\alpha_{T-ik}} = \frac{\alpha_0}{\alpha_T}$ of the polynomial on the right-hand side of Eq. 13 is always greater than 0. For the remaining unknown term $\left(\sqrt{1-\alpha_{T-(i+1)k}^2} - \frac{\sqrt{1-\alpha_{T-ik}^2}\alpha_{T-(i+1)k}}{\alpha_{T-ik}}\right)\boldsymbol{\epsilon}_\theta(\mathbf{x}_{T-ik}+\Delta \mathbf{x}_{T-ik}, T-ik)$, which contributes to the final expansion polynomial, we primarily prove that its factor $\sqrt{1-\alpha_{T-(i+1)k}^2} - \frac{\sqrt{1-\alpha_{T-ik}^2}\alpha_{T-(i+1)k}}{\alpha_{T-ik}}$ reaches its maximum value at $i = 0$. In practice, $\sqrt{1-\alpha_t^2}$ in DDPM and DDIM can be regarded as a monotonically increasing function as $t$ approaches 1, satisfying $\sqrt{1-\alpha_t^2}^{\cdots} < 0$ ($\cdots$ denotes the second-order derivative with respect to $t$). Thus, the aforementioned factor can be rewritten as

$$\sqrt{1-\alpha_{T-(i+1)k}^2} - \frac{\sqrt{1-\alpha_{T-ik}^2}\alpha_{T-(i+1)k}}{\alpha_{T-ik}}$$
$$= k\sqrt{1-\alpha_{T-(i+1)k}^2}\sqrt{1-\alpha_{T-ik}^2}\frac{\partial(\sqrt{1-\alpha_{T-(i+1)k}^2} - \sqrt{1-\alpha_{T-ik}^2})/\partial t}{\alpha_{T-ik}}, \quad \text{s.t., } k \text{ is small.} \tag{13}$$

The factors $\sqrt{1-\alpha_{T-(i+1)k}^2}$, $\sqrt{1-\alpha_{T-ik}^2}$, $\frac{\partial(\alpha_{T-ik}-\alpha_{T-(i+1)k})}{\partial t}$, and $\frac{1}{\alpha_{T-ik}}$ all have the desirable characteristic of decreasing monotonically as $t$ approaches 0. Therefore, $\sqrt{1-\alpha_{T-(i+1)k}^2} - \frac{\sqrt{1-\alpha_{T-ik}^2}\alpha_{T-(i+1)k}}{\alpha_{T-ik}}$ must take its maximum value when $i = 0$. Through a simple calculation using the most popular noise schedule widely applied in the Stable Diffusion family, we obtain this value as 0.126. Since the noise estimation model $\boldsymbol{\epsilon}_\theta(\cdot, \cdot)$ is differentiable, the power-of-1 error of 0.126 can be further analyzed using a first-order Taylor expansion:

$$\boldsymbol{\epsilon}_\theta(\mathbf{x}_{T-ik} + \Delta \mathbf{x}_{T-ik}, T-ik)$$
$$= \boldsymbol{\epsilon}_\theta(\mathbf{x}_{T-ik}, T-ik) + \Delta \mathbf{x}_{T-ik}\frac{\partial \boldsymbol{\epsilon}_\theta(\mathbf{x}_{T-ik}, T-ik)}{\partial \mathbf{x}_{T-ik}} + \mathcal{O}(\Delta^2 \mathbf{x}_{T-ik}). \tag{14}$$

We abbreviate the first-order partial derivatives of $\boldsymbol{\epsilon}_\theta(\mathbf{x}, t)$ with respect to $\mathbf{x}$ as $\dot{\boldsymbol{\epsilon}}_\theta(\mathbf{x}, t)$, and eventually Eq. 13 can be transformed into:

$$\|\mathbf{x}_0 - \mathbf{x}_0'\| = \left\|\Delta \mathbf{x}_T\left[\frac{\alpha_0}{\alpha_T} + \mathbf{Z}(0)\mathbb{E}_{t\sim \mathbf{Unif}(0,k,\cdots,N)}\left[\frac{\alpha_0}{\alpha_{t-k}}\frac{\alpha_t}{\alpha_T}\dot{\boldsymbol{\epsilon}}_\theta(\mathbf{x}_t, t)\right] + \mathcal{O}(\mathbf{Z}(0)^2)\right]\right\|, \tag{15}$$

where $\mathbf{Z}(t) = \sqrt{1-\alpha_{T-(t+1)k}^2} - \frac{\sqrt{1-\alpha_{T-tk}^2}\alpha_{T-(t+1)k}}{\alpha_{T-tk}}$.

Of course, we can also derive the relative error from noises at any timestep $H$:

$$\|\mathbf{x}_0 - \mathbf{x}_0'\| = \left\|\Delta \mathbf{x}_H\left[\frac{\alpha_0}{\alpha_H} + \mathbf{Z}(0)\mathbb{E}_{m\sim \mathbf{Unif}(1,\cdots,H/k)}\left[\frac{\alpha_0}{\alpha_{m-k}}\frac{\alpha_m}{\alpha_H}\dot{\boldsymbol{\epsilon}}_\theta(\mathbf{x}_m, m)\right] + \mathcal{O}(\mathbf{Z}(0)^2)\right]\right\|, \tag{16}$$

**Remarks.** The factor $\mathbf{Z}(0) = \sqrt{1 - \alpha_{T-k}^2} - \frac{\sqrt{1 - \alpha_T^2}\alpha_{T-k}}{\alpha_T}$ in SD V1.5 (Rombach et al., 2022), SD V2.1 (Rombach et al., 2022), SD XL (Podell et al.), Modelscope-T2V (Wang et al., 2023), Animatediff (Guo et al., 2023), and Latte (Ma et al., 2024) can be calculated as 0.126, which is smaller than 1. This means that Theorem 3.1 can be applied to those diffusion models, but it does not apply to several specialized models like SD V3 (Esser et al., 2024), which uses Rectified flow (Liu et al., 2022) and velocity estimation models. In such cases, the factors $\frac{\alpha_0}{\alpha_T}$ and $\mathbf{Z}$ in Eq. 15 directly degenerate to 1, preventing the conclusion that $\frac{\mathbb{E}_{\mathbf{x}_0, \mathbf{x}_0'} \|\mathbf{x}_0 - \mathbf{x}_0'\|}{\mathbb{E}_{\mathbf{x}_T, \mathbf{x}_T'} \|\mathbf{x}_T - \mathbf{x}_T'\|}$ exceeds 1. $\square$

## C.3 Entropy Shifts during Reverse Sampling

This subsection further elaborates on Theorem 3.1. We utilize the Fokker-Planck equations to analyze the entropy shift during the sampling process of diffusion models. We also introduce the definition of the reverse ODE from Song et al. (2023b):

$$d\mathbf{x}_t = \left[ f(\mathbf{x}_t, t) - \frac{1}{2}g^2(t)\nabla_{\mathbf{x}} \log q_t(\mathbf{x}) \right], \qquad \text{\# Continuous Reverse ODE}$$

$$d\mathbf{x}_t = \alpha_t \mathbf{x}_0 + \sigma_t \boldsymbol{\omega}_t, \qquad \text{\# Adding Noise} \tag{17}$$

where $f(\cdot, t) : \mathbb{R}^d \to \mathbb{R}^d$ is a vector-valued function called the *drift* coefficient of $\mathbf{x}_t$, and $g(\cdot) : \mathbb{R} \to \mathbb{R}$ is a scalar function known as the *diffusion* coefficient of $\mathbf{x}_t$. Note that we use $q_t(\mathbf{x})$ instead of $p(\mathbf{x}_t)$ for simplicity. $\boldsymbol{\omega}_t$ is the standard Wiener process. Then, the entropy shift from $q_1(\mathbf{x})$ to $q_0(\mathbf{x})$ can be defined as

$$\int q_0(\mathbf{x}) \log q_0(\mathbf{x}) d\mathbf{x} - \int q_1(\mathbf{x}) \log q_1(\mathbf{x}) d\mathbf{x}. \tag{18}$$

Continuing the derivation yields the following result:

$$\int \int_1^0 \frac{\partial}{\partial t} \left[ q_t(\mathbf{x}) \log q_t(\mathbf{x}) \right] dt d\mathbf{x}$$

$$= \int \int_1^0 \left[ \frac{\partial q_t(\mathbf{x})}{\partial t} \log q_t(\mathbf{x}) + \frac{\partial}{\partial t} q_t(\mathbf{x}) \right] dt d\mathbf{x}$$

$$= \int \int_1^0 \left[ \nabla_{\mathbf{x}} \cdot (h(\mathbf{x}_t, t) q_t(\mathbf{x}))(\log q_t(\mathbf{x}) + 1) \right] dt d\mathbf{x} \qquad \text{\# Fokker-Planck equations}$$

$$= \int \int_1^0 \left[ (\log q_t(\mathbf{x}) + 1)^{\mathrm{T}} (h(\mathbf{x}_t, t) q_t(\mathbf{x})) \right] dt d\mathbf{x} \qquad \text{\# Assumption 2}$$

$$= \int \int_1^0 \left[ (\log q_t(\mathbf{x}) + 1)^{\mathrm{T}} \left( \frac{d \log \alpha_t}{dt} \mathbf{x}_t - \frac{1}{2} \left[ \frac{d\sigma_t^2}{d_t} - 2\frac{d \log \alpha_t}{d_t} \sigma_t^2 \right] \nabla_{\mathbf{x}} \log q_t(\mathbf{x}) \right) \right] dt d\mathbf{x}$$

$$= \int_1^0 \int \frac{1}{2} - \left[ \frac{d\sigma_t^2}{d_t} - 2\frac{d \log \alpha_t}{d_t} \sigma_t^2 \right] \nabla_{\mathbf{x}} \log q_t(\mathbf{x}) d\mathbf{x} dt + \int_1^0 \int \frac{d \log \alpha_t}{dt} \mathbf{x}_t \log q_t(x) d\mathbf{x} dt$$

$$+ \int_1^0 \frac{d \log \alpha_t}{dt} \alpha_t^2 \frac{\mathbb{E}_{\mathbf{x}_0 \sim q_0(\mathbf{x})}[\mathbf{x}_0^2]}{2} dt + \int_1^0 \frac{d \log \alpha_t}{2dt} \sigma_t^2 dt - \frac{1}{4}\int_1^0 \left[ \frac{d\sigma_t^2}{d_t} - 2\frac{d \log \alpha_t}{d_t} \sigma_t^2 \right] [\log q_t(x)]^2 dt. \tag{19}$$

Substitute $\alpha_t = \exp\left(-\frac{1}{4}at^2 - \frac{1}{2}bt\right)$ and $\sigma_t = \sqrt{1 - \exp(-\frac{1}{2}at^2 - bt)}$ with $a = 19.9$ and $b = 0.1$ (Liu et al., 2022; Shao et al., 2024), we can obtain that

$$\int_1^0 \frac{d \log \alpha_t}{dt} \alpha_t^2 \frac{\mathbb{E}_{\mathbf{x}_0 \sim q_0(\mathbf{x})}[\mathbf{x}_0^2]}{2} dt$$

$$= \left[ -\frac{e^{-\frac{at^2}{2} - bt}}{4} + C \right] \Bigg|_1^0 \mathbb{E}_{\mathbf{x}_0 \sim q_0(\mathbf{x})}[\mathbf{x}_0^2]. \tag{20}$$

Obviously, the above result is a negative value. In contrast, when the noise schedule is Rectified flow (Liu et al., 2022) (applied in Stable Diffusion V3), the upper value $\int_1^0 \frac{d \log \alpha_t}{dt} \alpha_t^2 \frac{\mathbb{E}_{\mathbf{x}_0 \sim q_0(\mathbf{x})}[\mathbf{x}_0^2]}{2} dt$

is $(-(0 - \frac{0^2}{2}) + (1 - \frac{1^2}{2}))\frac{\mathbb{E}_{\mathbf{x}_0 \sim q_0(\mathbf{x})}[\mathbf{x}_0^2]}{2} = \frac{\mathbb{E}_{\mathbf{x}_0 \sim q_0(\mathbf{x})}[\mathbf{x}_0^2]}{4}$. Thus, from a noise schedule perspective, Rectified flow through sampling causes a relative increase in entropy, but DDPM/DDIM causes a relative decrease in entropy.

## C.4 INTRINSIC PRINCIPLES OF ONE STEP FORWARD-INVERSION

In this subsection, we present the proof of Theorem 3.2:

*Proof.* One step forward-inversion represents one additional step forward sampling and one step reverse sampling against the initial Gaussian noise, which can be denoted as

$$\mathbf{x}'_T = f_s(\mathbf{x}_T) = \textbf{DDIM-Inversion}(\textbf{DDIM}(\mathbf{x}_T)), \tag{21}$$

where **DDIM-Inversion**$(\cdot)$ refers to the sampling algorithm in Eq. 8 satisfy $t \geq s$. We can rewrite it in forms of differential equations:

$$\mathbf{x}'_T = \mathbf{x}_T - \frac{1}{\eta}\int_1^{1-\eta}\left[\frac{g^2(1)}{2}\nabla_{\mathbf{x}}\log q_1(\mathbf{x}) - \frac{g^2(1-\eta)}{2}\nabla_{\mathbf{x}}\log q_{1-\eta}(\mathbf{x})\right]dt, \tag{22}$$

where $1 - \eta$ denotes the next sampling moment when the timestep is 1. Adopting the classifier-free guidance paradigm (Ho & Salimans, 2021), Eq. 21 is rewritten as

$$\mathbf{x}'_T = \mathbf{x}_T - \frac{1}{\eta}\int_1^{1-\eta}\left[\frac{g^2(1)}{2}(\omega_1 + 1)\nabla_{\mathbf{x}}\log q_1(\mathbf{x}|\mathbf{c}) - \frac{g^2(1)}{2}\omega_1\nabla_{\mathbf{x}}\log q_1(\mathbf{x})\right.$$
$$\left. - \frac{g^2(1-\eta)}{2}(\omega_2 + 1)\nabla_{\mathbf{x}}\log q_{1-\eta}(\mathbf{x}|\mathbf{c}) + \frac{g^2(1-\eta)}{2}\omega_2\nabla_{\mathbf{x}}\log q_{1-\eta}(\mathbf{x})\right]dt, \tag{23}$$

where $\mathbf{c}$, $\omega_1$ and $\omega_2$ refer to the text prompt (*i.e.*, condition), the CFG scale at the timestep 1 and the CFG scale at the timestep $1 - \eta$, respectively. Given that 1 and $1 - \eta$ are very close and Assumption 2, we can use their midpoint to perform a Taylor expansion, thereby reducing the truncation error:

$$\mathbf{x}'_T = \mathbf{x}_T - \frac{1}{\eta}\int_1^{1-\eta}\frac{g^2(1-\frac{\eta}{2})}{2}\left[(\omega_1 - \omega_2)\nabla_{\mathbf{x}}\log q_{1-\frac{\eta}{2}}(\mathbf{x}|\mathbf{c}) - (\omega_1 - \omega_2)\nabla_{\mathbf{x}}\log q_{1-\frac{\eta}{2}}(\mathbf{x}) + \mathcal{O}(\frac{\eta}{2})\right]dt,$$
$$\implies d\mathbf{x} = -\frac{g^2(1-\frac{\eta}{2})}{2}\left[(\omega_1 - \omega_2)\nabla_{\mathbf{x}}\log q_{1-\frac{\eta}{2}}(\mathbf{c}|\mathbf{x}) + \mathcal{O}(\frac{\eta}{2})\right]dt. \tag{24}$$

With Eq. 24, it is possible to inject semantic information within the future timestep (*i.e.*, $t = 1 - \eta/2$) into the initial Gaussian noise (*i.e.*, $t = 1$) when there is a gap between $\omega_1$ and $\omega_2$.

$\square$

## C.5 THEORETICAL ANALYSIS OF SMOOTHING INITIALIZATION

**How Smooth Initialization Ensures Smoothness?**  Here we present the theoretical proof of Theorem 3.3. The proof closely follows the structure of Cohen et al. (2019)'s proof of Cohen et al. (2019, Theorem 1), with specific modifications to align with the smoothing initialization framework.

*Proof.* We begin the derivation by recapitulating one crucial lemma:

**Lemma C.2.** *(copy from Cohen et al. (2019, Lemma 4), Neyman-Pearson for Gaussians with different means) Let $X \sim \mathcal{N}(x, \sigma^2\mathbf{I})$ and $Y \sim \mathcal{N}(x + \delta, \sigma^2\mathbf{I})$. Let $h : \mathbb{R}^d \to \{0, 1\}$ be any deterministic or random function. Then:*

*1. If $S = \{z \in \mathbb{R}^d : \delta^T z \leq \beta\}$ for some $\beta$ and $\mathbb{P}(h(X) = 1) \geq \mathbb{P}(X \in S)$, then $\mathbb{P}(h(Y) = 1) \geq \mathbb{P}(Y \in S)$.*

*2. If $S = \{z \in \mathbb{R}^d : \delta^T z \geq \beta\}$ for some $\beta$ and $\mathbb{P}(h(X) = 1) \leq \mathbb{P}(X \in S)$, then $\mathbb{P}(h(Y) = 1) \leq \mathbb{P}(Y \in S)$.*

The contribution of this lemma is to demonstrate that the inputs will maintain consistent classification results across a range of perturbations. Unlike Cohen et al. (2019, Theorem 1), which directly focuses on the classification task, the robustness proof for SMOOTHINIT requires transforming the continuous regression task (*i.e.*, generative a better initial noise) into a discrete classification task. Thus, we define a deterministic function:

$$h(x) = \mathbb{1}\left[\|f_s(\mathbf{x}) - \epsilon^*\| \le \gamma\right]. \tag{25}$$

We additionally declare that $\epsilon^* \in \mathbb{R}^d$ and $\gamma \in \mathbb{R}^+$ are always existed in this case. Specifically, when $\epsilon^*$ is closen as $f_s(\mathbf{x})$, then $h(x) \equiv 1$. Conversely, when $\epsilon^* = f_s(\mathbf{x}) + \gamma + 1$, then $h(x) \equiv 0$. Given the two conditions above and the fact that the norm $\|f_s(\mathbf{x}) - \epsilon^*\|$ is first-order derivable with respect to $\epsilon^*$, the pair $(\epsilon^*, \gamma)$ satisfying the constraints must exist.

The next step is to solve a constraint on $\delta$ such that $\mathbb{1}\left[\|[g(\mathbf{x} + \delta) - \epsilon^*\| \le \gamma\right] = 1$ holds. A more intuitive goal is

$$\mathbb{P}(\mathbb{1}\left[\|f_s(\mathbf{x} + \epsilon + \delta) - \epsilon^*\| \le \gamma\right] = 1) \ge \mathbb{P}(\mathbb{1}\left[\|f_s(\mathbf{x} + \epsilon + \delta) - \epsilon^*\| \le \gamma\right] = 0). \tag{26}$$

We can use Lemma C.2 by defining two special half-spaces:

$$\Phi\left(\frac{(\mathbf{z} - \mathbf{x})}{\sigma_{\text{sm}}}\right) = \underline{p_1}$$
$$\implies A = \{\mathbf{z} : \delta^T(\mathbf{z} - \mathbf{x}) \le \sigma_{\text{sm}}\|\delta\|\Phi^{-1}(\underline{p_1})\}$$

$$\Phi\left(\frac{(\mathbf{z} - \mathbf{x})}{\sigma_{\text{sm}}}\right) = \overline{p_0} \tag{27}$$
$$\implies B = \{\mathbf{z} : \delta^T(\mathbf{z} - \mathbf{x}) \ge \sigma_{\text{sm}}\|\delta\|\Phi^{-1}(\overline{p_0})\}$$

Then we can obtain

$$\mathbb{P}(\mathbb{1}\left[\|f_s(\mathbf{x} + \epsilon + \delta) - \epsilon^*\| \le \gamma\right] = 1) \ge \mathbb{P}(\mathbf{x} + \epsilon + \delta \in A)$$
$$> \mathbb{P}(\mathbf{x} + \epsilon + \delta \in B) \ge \mathbb{P}(\mathbb{1}\left[\|f_s(\mathbf{x} + \epsilon + \delta) - \epsilon^*\| \le \gamma\right] = 0). \tag{28}$$

The derivation of the above two equations leads to

$$\Phi\left(\Phi^{-1}(\underline{p_1}) - \frac{\|\delta\|}{\sigma_{\text{sm}}}\right) = \mathbb{P}(\mathbf{x} + \epsilon + \delta \in A)$$
$$\implies \mathbb{P}(\mathbf{x} + \epsilon + \delta \in A) = \Phi(\Phi^{-1}(\underline{p_1}) - \frac{\|\delta\|}{\sigma_{\text{sm}}})$$

$$\tag{29}$$

$$\Phi\left(\Phi^{-1}(\overline{p_0}) + \frac{\|\delta\|}{\sigma_{\text{sm}}}\right) = \mathbb{P}(\mathbf{x} + \epsilon + \delta \in B)$$
$$\implies \mathbb{P}(\mathbf{x} + \epsilon + \delta \in B) = \Phi(\Phi^{-1}(\overline{p_0}) + \frac{\|\delta\|}{\sigma_{\text{sm}}})$$

In order to ensure that Eq. 28 holds, it is necessary to ensure that:

$$\Phi(\Phi^{-1}(\underline{p_1}) - \frac{\|\delta\|}{\sigma_{\text{sm}}}) \ge \Phi(\Phi^{-1}(\overline{p_0}) + \frac{\|\delta\|}{\sigma_{\text{sm}}})$$
$$\implies \|\delta\| \le \frac{\sigma_{\text{sm}}}{2}\left[\Phi^{-1}(\underline{p_1}) - \Phi^{-1}(\overline{p_0})\right]. \tag{30}$$

The proof is complete. $\qquad\square$

**The Nature of Smooth Initialization.**

*Proof.* In this section, we derive Corollary 3.4 by transforming the ordinary differential equation (Eq. 24) into its discrete form:

$$f_s(\mathbf{x}_T) = \mathbf{x}_T - \frac{\sigma_{T-\frac{1}{2}}}{\alpha_{T-\frac{1}{2}}} \frac{\alpha_{T-1}\sigma_T - \alpha_T\sigma_{T-1}}{\alpha_{T-1}}(\omega_1 - \omega_2)\frac{\partial\left[-\log q_{1-\frac{\eta}{2}}(\mathbf{x})\right]}{\partial \mathbf{x}_0}, \qquad (31)$$

where $\mathbf{x}_T$ and $\mathbf{x}_0$ denote the Gaussian noise and the "clean" video. This theoretical conclusion can be directly obtained from the definitions of DDIM and DDIM-Inversion. It is important to note that Eq. 31 also includes the transformation of the vanilla score function $\nabla_{\mathbf{x}} \log q_{1-\frac{\eta}{2}}(\mathbf{x}) = \frac{\partial \log q_{1-\frac{\eta}{2}}(\mathbf{x})}{\partial x_{1-\frac{\eta}{2}}}$ into $-\frac{1}{\alpha_{1-\frac{\eta}{2}}}\frac{\partial \log q_{1-\frac{\eta}{2}}(\mathbf{x})}{\partial x_0}$. This is done with the following lemma:

**Lemma C.3.**

$$\frac{\partial \log p(\mathbf{x}_t|\mathbf{x}_0)}{\partial \mathbf{x}_0} = \frac{\partial \log \exp(\frac{\|\mathbf{x}_t - \alpha_t\mathbf{x}_0\|}{2\sigma_t^2})}{\partial \mathbf{x}_0} = -\alpha_t\frac{\alpha_t\mathbf{x}_0 - \mathbf{x}_t}{\sigma_t^2} = \frac{\alpha_t\boldsymbol{\epsilon}_\theta(\mathbf{x}_t, t)}{\sigma_t},$$

$$\nabla_{\mathbf{x}} \log q_t(\mathbf{x}) = \frac{\partial \log p(\mathbf{x}_t|\mathbf{x}_0)}{\partial \mathbf{x}_t} = \frac{\partial \log \exp(\frac{\|\mathbf{x}_t - \alpha_t\mathbf{x}_0\|}{2\sigma_t^2})}{\partial \mathbf{x}_t} = -\frac{\mathbf{x}_t - \alpha_t\mathbf{x}_0}{\sigma_t^2} = -\frac{\boldsymbol{\epsilon}_\theta(\mathbf{x}_t, t)}{\sigma_t}, \qquad (32)$$

*Then,* $\frac{\partial \log p(\mathbf{x}_t|\mathbf{x}_0)}{\partial \mathbf{x}_0} = -\alpha_t\nabla_{\mathbf{x}} \log q_t(\mathbf{x}).$

For ease of derivation, we first adopt the FORWARD-INVERSION operator in the smoothing function as $f_s(\cdot)$, then

$$\mathbb{E}_{\epsilon\sim\mathcal{N}(0,\sigma_{\text{sm}}^2\mathbf{I})}[f_s(\mathbf{x} + \epsilon)]$$

$$= \mathbb{E}_{\epsilon\sim\mathcal{N}(0,\sigma_{\text{sm}}^2\mathbf{I})}f_s(\mathbf{x}) + \mathbb{E}_{\epsilon\sim\mathcal{N}(0,\sigma_{\text{sm}}^2\mathbf{I})}[\epsilon]^{\mathrm{T}}\nabla_{\mathbf{x}}f_s(\mathbf{x}) + \frac{1}{2}\mathbb{E}_{\epsilon\sim\mathcal{N}(0,\sigma_{\text{sm}}^2\mathbf{I})}[\epsilon]^{\mathrm{T}}\mathbf{H}[\epsilon] + \mathcal{O}(\epsilon^3)$$

$$= f_s(\mathbf{x}) + \frac{\sigma_{\text{sm}}^2}{2}\mathbb{E}_{\epsilon\sim\mathcal{N}(0,\mathbf{I})}[\epsilon]^{\mathrm{T}}\mathbf{H}[\epsilon] \qquad (33)$$

$$= f_s(\mathbf{x}) + \frac{\sigma_{\text{sm}}^2}{4}\mathbf{1}^T\times\mathbf{A}\odot\mathbf{H}\times\mathbf{1},$$

where $\mathbf{H}, \mathbf{1}, \times$ and $\odot$ stand for the Hessian matrix, the unit vector, the matrix multiplication and the Hadamard product, respectively. $\mathbf{A}$ is a matrix whose nondiagonal elements obey a unit Gaussian distribution $\mathcal{N}(0, \mathbf{I})$ and whose diagonal elements obey a chi-square distribution with $\mathbb{E}[\mathbf{A}_{\text{ii}}] = 1$. Given this, we can further get

$$f_s(\mathbf{x}) + \frac{\sigma_{\text{sm}}^2}{4}\mathbf{1}^T\times\mathbf{A}\odot\mathbf{H}\times\mathbf{1}$$

$$= f_s(\mathbf{x}) + \frac{\sigma_{\text{sm}}^2}{4}\text{tr}(\mathbf{A}\odot\mathbf{H}) + C_{\text{const}} \qquad (34)$$

$$= f_s(\mathbf{x}) + \frac{\sigma_{\text{sm}}^2}{4}\text{tr}(\mathbf{H}) + C_{\text{const}},$$

Substituting Eq. 31 into the above equation, we can obtain

$$f_s(\mathbf{x}_T) = \mathbf{x}_T - \frac{\sigma_{T-\frac{1}{2}}}{\alpha_{T-\frac{1}{2}}} \frac{\alpha_{T-1}\sigma_T - \alpha_T\sigma_{T-1}}{\alpha_{T-1}}(\omega_1 - \omega_2)\frac{\partial\left[-\log q_{1-\frac{\eta}{2}}(x)\right]}{\partial \mathbf{x}_0},$$

$$g(\mathbf{x}_T) = f_s(\mathbf{x}_T) - \frac{\sigma_{\text{sm}}^2\sigma_{T-\frac{1}{2}}}{4\alpha_{T-\frac{1}{2}}} \frac{\alpha_{T-1}\sigma_T - \alpha_T\sigma_{T-1}}{\alpha_{T-1}}(\omega_1 - \omega_2)\text{tr}\left(\frac{\partial^3\left[-\log q_{1-\frac{\eta}{2}}(x)\right]}{\partial \mathbf{x}_0^3}\right). \qquad (35)$$

The proof is complete. □

## C.6 THEORETICAL ANALYSIS OF FAST-SMOOTH INITIALIZATION

*Proof.* This subsection clarifies why FAST-SMOOTHINIT converges to the optimal initial noise by being transformed into the form of differential equations. First, we give the following iterative form:

$$\mathbf{x}_k = f_s(\mathbf{x}_{k-1} + \sigma_{\text{sm}}^{k-1}\epsilon_{k-1}), \text{ where } \epsilon_{k-1} \sim \mathcal{N}(0, \mathbf{I}) \text{ and } k \in \{1, \cdots, K\}. \quad (36)$$

This form is consistent with Algorithm 1 as outlined in the main paper. For the convenience of substitution, we modify Eq. 36 while keeping the logic of the algorithm consistent as follows

$$\mathbf{x}_k = f_s(\mathbf{x}_{k-1}, \sigma_{\text{sm}}^{k-2}\epsilon_{k-2}), \text{ s.t. } \epsilon_{k-2} \sim \mathcal{N}(0, \mathbf{I}) \text{ and } k \in \{2, \cdots, K\}$$
$$\implies \mathbf{y}_k = f_s(\mathbf{y}_{k-1}) + \sigma_{\text{sm}}^{k-1}\epsilon_{k-1}, \text{ s.t. } \mathbf{y}_k = \mathbf{x}_k + \sigma_{\text{sm}}^{k-1}\epsilon_{k-1} \text{ and } k \in \{1, \cdots, K\}. \quad (37)$$

In the limit as $K \to \infty$, the Markov chain $\{\mathbf{y}_k\}_{k=1}^K$ becomes a continuous stochastic process $\{\mathbf{y}(k)\}_{k=0}^1$. Similarly, $\{\sigma_{\text{sm}}^k\}_{k=1}^K$ becomes a function $\{\sigma_{\text{sm}}(k)\}_{k=0}^1$ and $\{\epsilon_k\}_{k=1}^K$ becomes a function $\{\epsilon(k)\}_{k=0}^1$, where we now use the continuous time variable $k \in [0, 1]$ for indexing, rather than the integer $k \in \{1, 2, \cdots, K\}$. Let $\Delta k = \frac{1}{K}$, we can rewrite Eq. 37 as follows with $k = \{0, \frac{1}{K}, \frac{2}{K}, \cdots, 1\}$:

$$\mathbf{y}(k + \Delta k) = f_s(\mathbf{y}(k)) + \sigma_{\text{sm}}(k)\epsilon(k)$$
$$\implies \mathbf{y}(k + \Delta k) - \mathbf{y}(k) = \left[-\frac{1}{\Delta k}\mathbf{y}(k) + \frac{1}{\Delta k}f_s(\mathbf{y}(k))\right]\Delta k + \frac{1}{\sqrt{\Delta k}}\sigma_{\text{sm}}(k)\sqrt{\Delta k}\epsilon(k)$$
$$\implies d\mathbf{y} = -\left[\frac{\sigma_{T-\frac{1}{2}}}{\Delta k}\frac{\alpha_{T-1}\sigma_T - \alpha_T\sigma_{T-1}}{\alpha_{T-1}}(\omega_1 - \omega_2)\right]\nabla_{\mathbf{y}}\log q_{1-\frac{\eta}{2}}(\mathbf{y}(k))dk + \left[\frac{1}{\sqrt{\Delta k}}\sigma_{\text{sm}}(k)\right]d\overline{\boldsymbol{\omega}}_k, \quad (38)$$

where $\overline{\boldsymbol{\omega}}_k$ is a standard Gaussian noise independent of $\mathbf{y}(k)$. However, because the score function is $\nabla_{\mathbf{y}}\log q_{1-\frac{\eta}{2}}(\mathbf{y}(k))$ rather than $\nabla_{\mathbf{y}}\log q_k(\mathbf{y}(k))$, the Fokker-Planck-Kolmogorov (FPK) equation does not transform the stochastic differential equation into an ordinary differential equation in an elegant manner. Unfortunately, $\nabla_{\mathbf{y}}\log q_k(\mathbf{y}(k))$ cannot be estimated since there is no corresponding optimization objective during training, so we directly use $\nabla_{\mathbf{y}}\log q_{1-\frac{\eta}{2}}(\mathbf{y}(k))$ to implement FAST-SMOOTHINIT. Given this, Eq. 38 can be rewritten as

$$d\mathbf{y} = -\left[\frac{\sigma_{T-\frac{1}{2}}}{\Delta k}\frac{\alpha_{T-1}\sigma_T - \alpha_T\sigma_{T-1}}{\alpha_{T-1}}(\omega_1 - \omega_2)\right]\nabla_{\mathbf{y}}\log q_k(\mathbf{y}(k))dk + \left[\frac{1}{\sqrt{\Delta k}}\sigma_{\text{sm}}(k)\right]d\overline{\boldsymbol{\omega}}_k$$
$$\implies d\mathbf{y} = \left[-\frac{\sigma_{T-\frac{1}{2}}}{\Delta k}\frac{\alpha_{T-1}\sigma_T - \alpha_T\sigma_{T-1}}{\alpha_{T-1}}(\omega_1 - \omega_2) + \frac{1}{2\Delta k}\sigma_{\text{sm}}^2(k)\right]\nabla_{\mathbf{y}}\log q_k(\mathbf{y}(k))dk. \quad (39)$$

Substituting the notation in Algorithm 1, we can obtain the following differential equation:

$$d\mathbf{x} = \left[-\frac{\sigma_{T-\frac{1}{2}}}{\Delta t}\frac{\alpha_{T-1}\sigma_T - \alpha_T\sigma_{T-1}}{\alpha_{T-1}}(\omega_1 - \omega_2) + \frac{1}{2\Delta t}\sigma_{\text{sm}}^2(1 - \gamma_{\text{decay}})^{2i}\right]\nabla_{\mathbf{x}}\log q_t(\mathbf{x})dt. \quad (40)$$

The proof is complete. □

The ordinary differential equation in Eq. 39 can be solved analytically for $k$ within a specified interval. The best approximation to reduce truncation error is then achieved using a Taylor expansion, similar to several popular ODE solvers (Lu et al., 2022c;b; Zhang & Chen, 2022). are essentially momentum-based, meaning they are implemented by making the new $\mathbf{x}_{k-1}$ in Eq. 36 a weighted average of $\mathbf{x}_{k-1}$, $\mathbf{x}_{k-2}$ and $\mathbf{x}_{k-3}$. Then the algorithm can be transferred to

$$
\begin{aligned}
&\textbf{Input:} \quad \mathbf{x}_0,\ f_s(\cdot),\ \gamma_0,\ \gamma_1,\ \gamma_2,\ \{\sigma_{\mathrm{sm}}^k\}_{k=0}^{K-1} \\
&\mathbf{m}_0 \leftarrow \mathbf{x}_0 \\
&\mathbf{m}_1 \leftarrow \mathbf{x}_0 \\
&\mathbf{m}_2 \leftarrow \mathbf{x}_0 \\
&k \leftarrow 0 \\
&\textbf{While } k \leq K \\
&\quad \textbf{do} \\
&\qquad \mathbf{x}_k^{\mathrm{tmp}} \leftarrow \gamma_0 \mathbf{m}_0 + \gamma_1 \mathbf{m}_1 + \gamma_2 \mathbf{m}_2 \\
&\qquad \mathbf{x}_{k+1} \leftarrow f_s(\mathbf{x}_k^{\mathrm{tmp}} + \sigma_{\mathrm{sm}}^k \epsilon_k) \\
&\qquad \mathbf{m}_2 \leftarrow \mathbf{m}_1 \\
&\qquad \mathbf{m}_1 \leftarrow \mathbf{m}_0 \\
&\qquad \mathbf{m}_0 \leftarrow \mathbf{x}_{k+1} \\
&\quad \textbf{done} \\
&\textbf{Output:} \quad \mathbf{x}_K,
\end{aligned}
\tag{41}
$$

where $\gamma_0$, $\gamma_1$ and $\gamma_2$ are weighted averages satisfying a sum of $1$. Moreover, the pipeline in Eq. 41 is essentially the same as the algorithm presented in Algorithm 1. Motivated by Zheng et al. (2023) that analytical solutions may not be optimal in practical applications, we decided to treat $\gamma_0$, $\gamma_1$ and $\gamma_2$ as tunable parameters.

Thus, FAST-SMOOTHINIT is an algorithm similar to DPM-Solver (Lu et al., 2022c) and DPM-Solver++ (Lu et al., 2022b) which indeed reduces truncation errors through the momentum mechanism. The proof is complete.

## D  RELATED WORK

In this section, we discuss a series of plug-and-play algorithms focused on VDMs, including FREEINIT (Wu et al., 2023), FREENOISE (Qiu et al., 2024), UNICTRL (Chen et al., 2024b), and I4VGEN (Guo et al., 2024). FREEINIT primarily uses DDIM (Song et al., 2023a) and the diffusion forward process to generate new noise and reinitialize the noise by blending low-frequency components with high-frequency noise using a spatio-temporal filter, ultimately synthesizing the "clean" video through DDIM sampling. FREENOISE is a training-free approach for synthesizing longer videos, ensuring both high video quality and computational efficiency. UNICTRL maintains semantic consistency across frames using cross-frame self-attention control, while simultaneously enhancing motion quality and spatiotemporal consistency through motion injection and spatiotemporal synchronization. I4VGEN first generates high-quality images using a T2I diffusion model, then transforms them into Gaussian noise with preserved semantic information through a standard diffusion forward process. Next, it incorporates the VDM's temporal information using the score distillation sampling (SDS) algorithm, and finally samples from the standard DDIM. Since the official implementation[4] of I4VGEN lacked the necessary packages and code as of press time, it is not compared in this paper.

## E  ADDITIONAL EXPERIMENTS

Here, we present a series of additional experiments to further validate some important insights outlined in the main paper.

### E.1  IMAGE SYNTHESIS VS. VIDEO SYNTHESIS WITH DPM-SOLVER

We provide an example to illustrate the phenomenon "several significant training-free sampling methods in video synthesis do not perform as well as in image synthesis;' outlined in Sec. 1, using

---

[4]https://github.com/xiefan-guo/i4vgen

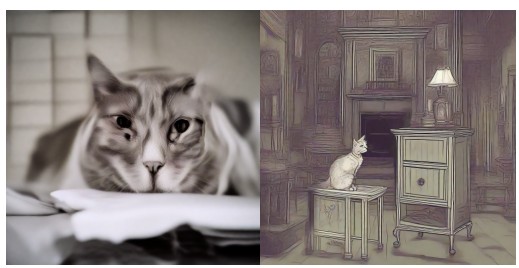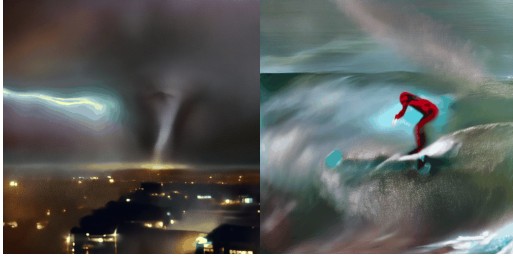

**DPM-Solver in Image Synthesis**          **DPM-Solver in Video Synthesis (Random Frame)**

Figure 8: The example of "DPM-Solver (NFE=5) performs less effectively in video synthesis compared to image synthesis".

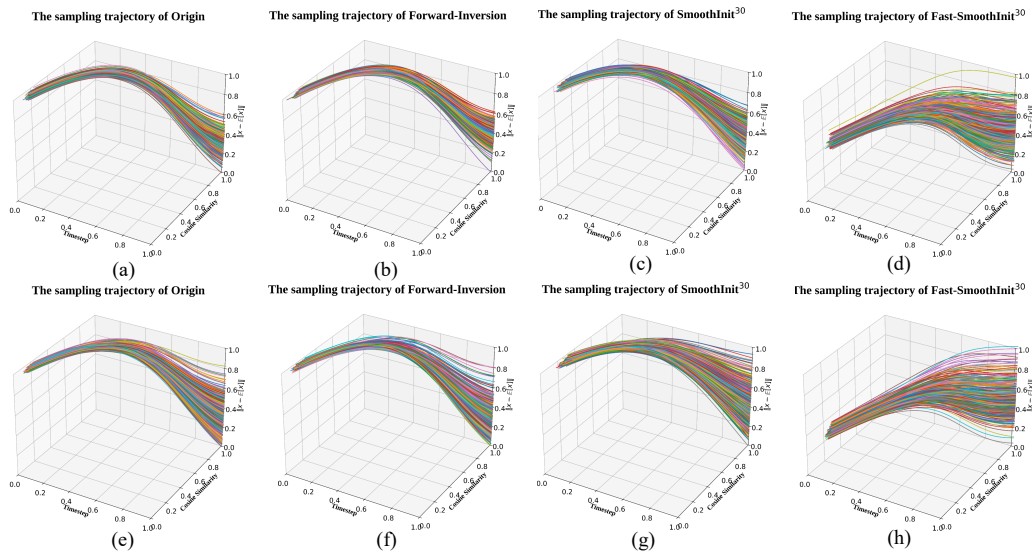

Figure 9: Visualization of sampling trajectories in MODELSCOPE-T2V. The prompts for (a), (b), (c), and (d) are "A cat wearing sunglasses and working as a lifeguard at a pool", and for (e), (f), (g), and (h) are "Spiderman is surfing". Each subgraph visualizes 500 sampling trajectories.

DPM-Solver with the number of function evaluations (NFE) set to 5. The visualization outcomes are presented in Fig. 8. We observe that the quality of image frames generated by ANIMATEDIFF (Guo et al., 2023) is worse compared to those generated by SD XL (Podell et al.), due to unnatural dynamics and the use of low-quality datasets for open-source video model training.

### E.2    3D SAMPLING TRAJECTORY VISUALIZATION

We present the 3D sampling trajectory of two prompts "A cat wearing sunglasses and working as a lifeguard at a pool" and "Spiderman is surfing" in Fig. 9. Different from the 2D sampling trajectory presented in the main paper, this part of the visualization is relatively more information-intensive, and the visualizations obtained from the two prompts are essentially the same.

### E.3    SMOOTHING INITIALIZATION MEETS DPM-SOLVER++

We similarly conduct experiments using a different scheduler, DPM-Solver++ (Lu et al., 2022b). The results in Table 6 show that SMOOTHINIT and FAST-SMOOTHINIT continue to perform well, with significant improvements across all metrics except for the metric GPT4O-SCORE. We hypothesize that the decrease in GPT4o-Score may be due to SMOOTHINIT and FAST-SMOOTHINIT reducing the temporal variability of the synthesized video with DPM-Solver++.

Table 6: Quantitative comparison after replacing DDIM with DPM-Solver++. The experiments were performed on ANIMATEDIFF (SD V1.5, MOTION ADAPTER V3) and sampled 15 steps.

| METHOD | UMT-FVD ($\downarrow$) | MTSCORE ($\uparrow$) | UMTSCORE ($\uparrow$) | GPT4O-MTSCORE ($\uparrow$) |
|---|---|---|---|---|
| ORIGIN | 244.84 | 0.4759 | 2.73 | 2.84 |
| FORWARD-INVERSION | 245.49 | 0.4670 | 2.77 | 2.82 |
| SMOOTHING[30] | 243.84 | 0.4761 | 2.80 | 2.82 |
| FAST-SMOOTHING[10] | 243.83 | 0.4788 | 2.75 | 2.56 |
| FAST-SMOOTHING[30] | 241.41 | 0.4850 | 2.90 | 2.68 |

## E.4 ADDITIONAL EXPERIMENTS OF TRUNCATED GAUSSIAN NOISE VS. STANDARD GAUSSIAN NOISE

In Table 5 presented in the main paper, we substantiate that uniform noise performs better than Gaussian noise. However, a remaining question is whether this improvement arises from the removal of some outliers of Gaussian noise from the mean. To investigate this, we further compare the performance of truncated Gaussian noise and standard Gaussian noise on SMOOTHINIT. The results in Table 7 demonstrate that truncated Gaussian noise performs worse than standard Gaussian noise. Therefore, the superior effectiveness of uniform noise can be attributed to the properties of its probability density function.

Table 7: Ablation studies between Gaussian noise and Truncated Gaussian noise for initialization.

| ITERATION NUMBER | NOISE TYPE | $\sigma_{sm}$ | UMT-FVD ($\downarrow$) | | UMTSCORE ($\uparrow$) | |
|---|---|---|---|---|---|---|
| ($\omega_1 : \omega_2$) | N/A | N/A | (7.5:7.5) | (7.5:1) | (7.5:7.5) | (7.5:1) |
| | Gaussian | 0.100 | 270.91 | 272.53 | 2.93 | 2.96 |
| | Gaussian | 0.075 | 276.76 | 267.45 | 2.93 | 3.07 |
| | Gaussian | 0.050 | 269.28 | 271.40 | 2.91 | 2.98 |
| 30 | Gaussian | 0.025 | 267.29 | 266.55 | 2.94 | 3.02 |
| | Truncated Gaussian | 0.100 | 268.15 | 269.72 | 2.94 | 2.92 |
| | Truncated Gaussian | 0.075 | 262.76 | 262.84 | 2.90 | 2.97 |
| | Truncated Gaussian | 0.050 | 281.57 | 261.15 | 2.83 | 3.05 |
| | Truncated Gaussian | 0.025 | 266.09 | 265.85 | 2.97 | 3.00 |
| | Gaussian | 0.100 | 262.14 | 264.09 | 2.98 | 3.02 |
| | Gaussian | 0.075 | 274.78 | 267.15 | 2.89 | 3.00 |
| | Gaussian | 0.050 | 265.16 | 266.80 | 3.00 | 2.97 |
| 10 | Gaussian | 0.025 | 268.73 | 265.20 | 2.93 | 2.97 |
| | Truncated Gaussian | 0.100 | 268.15 | 266.37 | 2.94 | 2.92 |
| | Truncated Gaussian | 0.075 | 262.76 | 262.84 | 2.90 | 2.97 |
| | Truncated Gaussian | 0.050 | 281.57 | 261.15 | 2.83 | 3.05 |
| | Truncated Gaussian | 0.025 | 266.09 | 265.85 | 2.97 | 3.00 |

## E.5 ADDITIONAL EXPERIMENTS OF FAST-SMOOTHINIT

We present the complete results of the ablation studies on $\gamma_{decay}$ and $\gamma_m$ in Fig. 10. As shown, FAST-SMOOTHINIT[30] generally performs better than FAST-SMOOTHINIT[10], particularly on UMT-FVD. Additionally, the optimal $\gamma_{decay}$ and $\gamma_m$ settings differ between FAST-SMOOTHINIT[10] and FAST-SMOOTHINIT[30]. The optimal configurations, ($\gamma_{decay} = 0.05, \gamma_m = 0.05$) and ($\gamma_{decay} = 0.25, \gamma_m = 0.01$) for FAST-SMOOTHINIT[10] and FAST-SMOOTHINIT[30], are presented in the main paper.

## E.6 ADDITIONAL EXPERIMENTS OF DIFFUSION TRANSFORMER

We also conduct experiments on LATTE (Ma et al., 2024), a VDM based on the diffusion transformer (DiT) architecture. We find that the videos generated by LATTE exhibit less variability compared to those generated by other VDMs. Specifically, the frames within LATTE's synthesized videos remain nearly identical. As shown in Table 8, we also observe that SMOOTHINIT and FAST-SMOOTHINIT are less effective on LATTE than on other models, likely due to the specificity of the DiT architecture. However, both methods improve frame-to-frame variability, as demonstrated by the amplitude-dependent metrics GPT4O-MTSCORE and MTSCORE (Yuan et al., 2024).

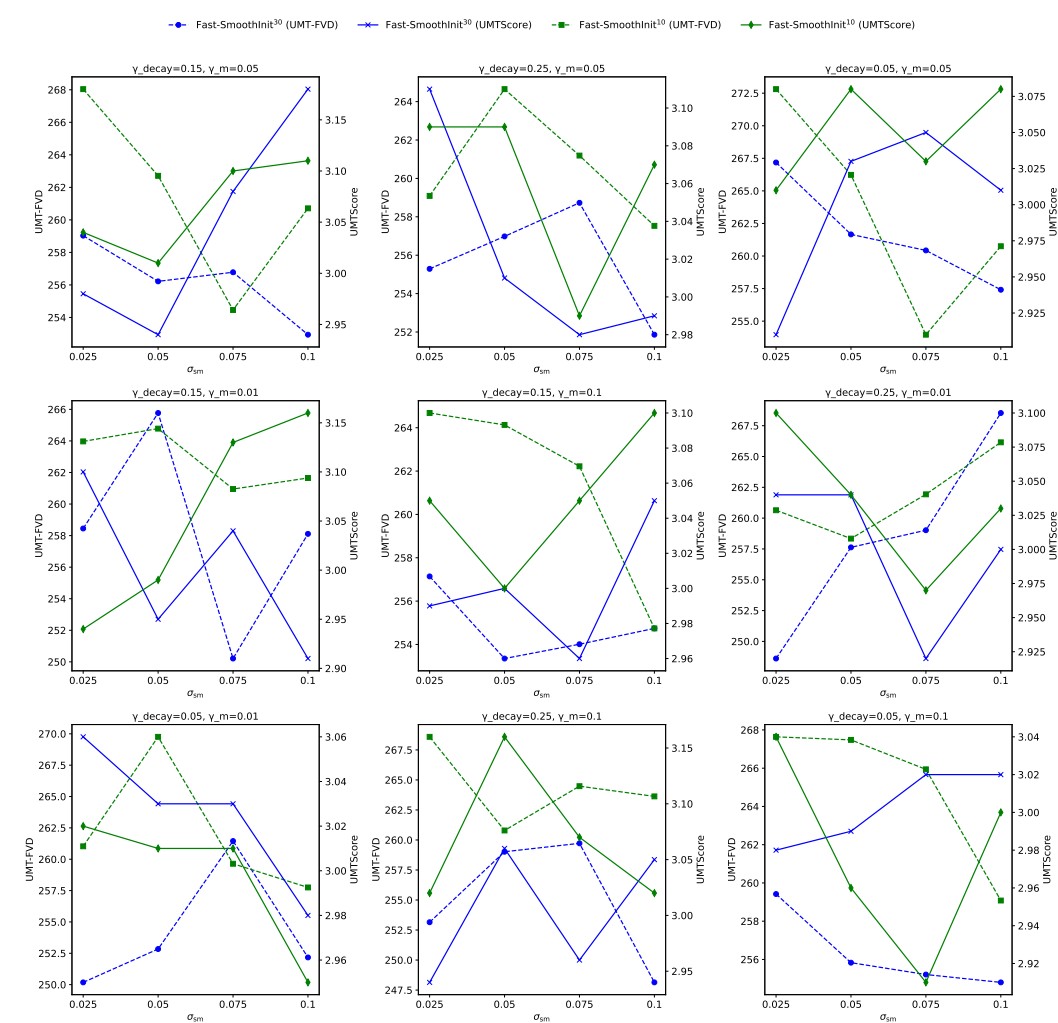

Figure 10: Ablation studies of $\sigma_{sm}$ and $\gamma_m$ within FAST-SMOOTHINIT. We present only the optimal hyperparameter settings in the main paper, while the complete ablation experiments with their corresponding hyperparameter settings are provided here.

Table 8: Quantitative comparison on LATTE (*w.r.t.*, DiT). Both FAST-SMOOTHINIT[10] and FAST-SMOOTHINIT[30] were performed with $\gamma_m = 0.95$

| METHOD | UMT-FVD ($\downarrow$) | MTSCORE ($\uparrow$) | UMTSCORE ($\uparrow$) | GPT4O-MTSCORE ($\uparrow$) |
|---|---|---|---|---|
| ORIGIN | 213.42 | 0.3854 | 2.65 | 2.23 |
| BASELINE (V2) | 216.59 | 0.3834 | 2.59 | 2.26 |
| ENSEMBLE (30, 0.025, UNI) | 220.00 | 0.3947 | 2.54 | 2.38 |
| FAST-SMOOTHING (10) | 220.64 | 0.3861 | 2.62 | 2.39 |
| FAST-SMOOTHING (30) | 219.63 | 0.3948 | 2.51 | 2.31 |

### E.7 ADDITIONAL ABLATION STUDIES OF $\omega_1 : \omega_2 = 7.5 : 7.5$

We present additional ablation studies on the hyperparameter noise level $\sigma_{sm}$ and the iteration number $K$ for the case where $\omega_1 : \omega_2 = 7.5 : 7.5$. As established in Theorem 3.2, injecting semantic information into the initial Gaussian noise is essentially impossible when $\omega_1$ equals $\omega_2$. This is further supported by the observation that the best performance in Fig. 11 is inferior to that in Fig. 4.

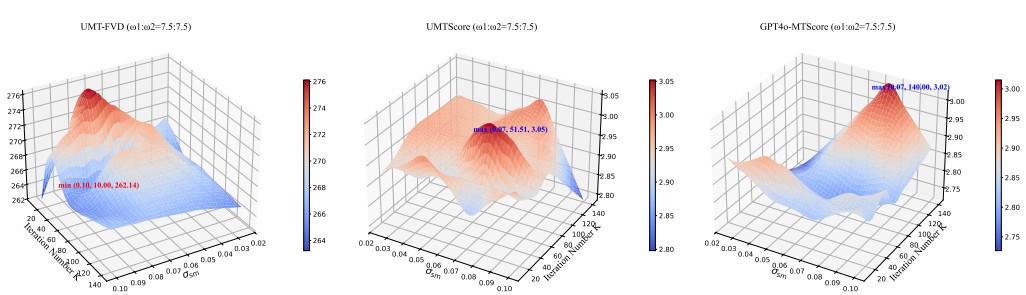

Figure 11: Ablation Studies on hyperparameter the noise level $\sigma_{\text{sm}}$ and the iteration number $K$. Compared to Fig. 4 in the main paper, the regularity in this figure is relatively low. This is because the semantic information was not successfully injected when using the ratio $\omega_1 : \omega_2 = 7.5 : 7.5$.

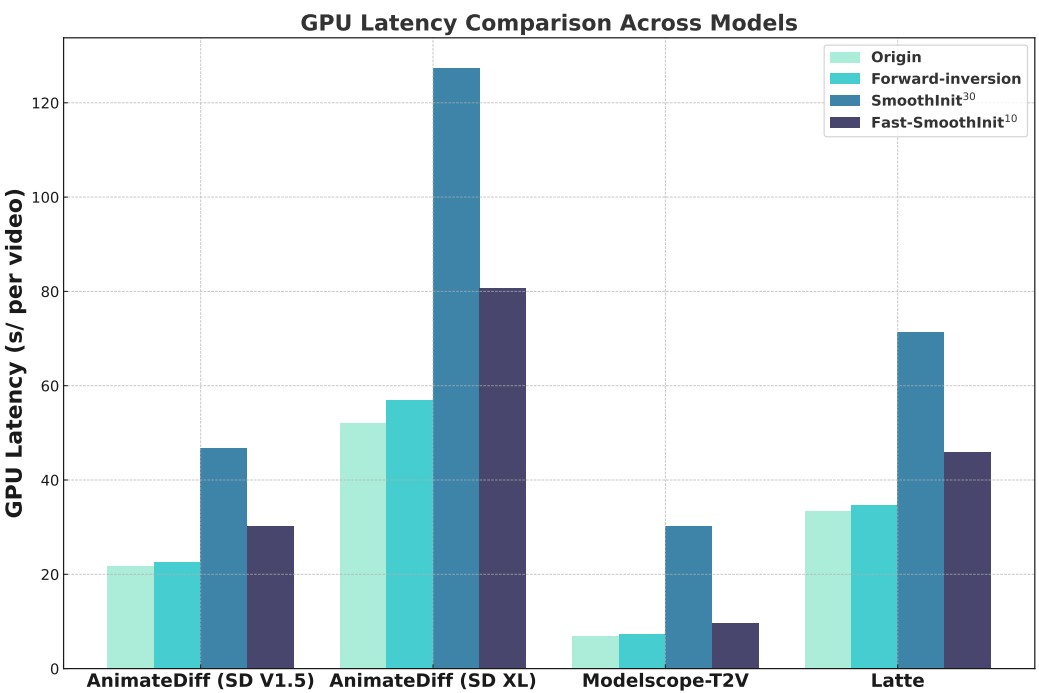

Figure 12: The GPU latency comparison between our proposed methods and baseline.

### E.8 GPU LATENCY ANALYSIS

SMOOTHINIT and FAST-SMOOTHINIT are two training-free algorithms designed to improve composite image quality by optimizing the initial Gaussian noise. However, since the optimization step requires repeated execution of **DDIM-Inversion**(**DDIM**($\cdot$)), this inevitably adds computational overhead. Fig. 12 presents the actual GPU latency on a single RTX 4090. While SMOOTHINIT and FAST-SMOOTHINIT introduce additional overhead, the need to compute text embeddings and other auxiliary variables before each reverse process ensures that even SMOOTHINIT[30] remains within acceptable limits.

## F SYNTHESIZED VIDEO VISUALIZATION

To reduce the size of the generated pdf, we downsample the video frames and present them here. We present the synthesized video visualization of ANIMATEDIFF (SD V1.5, MOTION ADAPTER V3) in Fig. 13-19, the synthesized video visualization of ANIMATEDIFF (SD XL, MOTION ADAPTER BETA) in Fig. 20-26 and the synthesized video visualization of MODELSCOPE-T2V in Fig. 27-33.

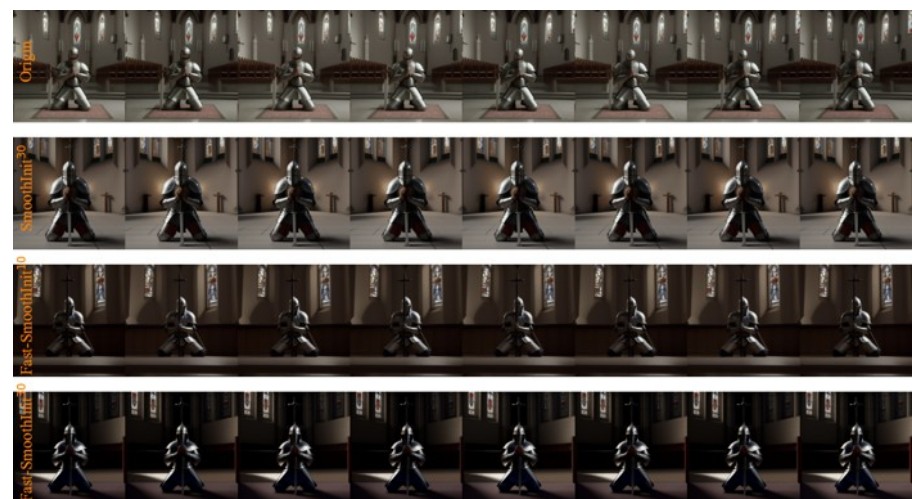

Figure 13: The synthesized video visualization of ANIMATEDIFF (SD V1.5, MOTION ADAPTER V3) with prompt "A knight kneeling in a chapel, his sword laid before him as he prays for strength before a great battle".

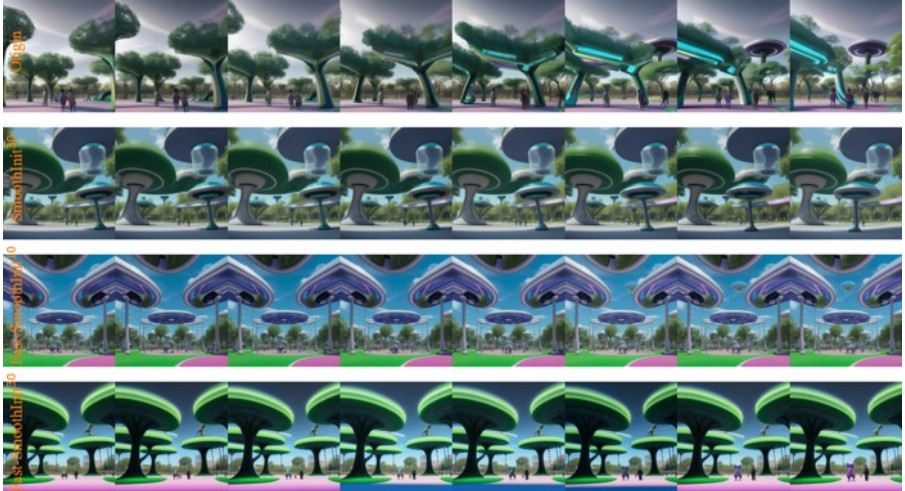

Figure 14: The synthesized video visualization of ANIMATEDIFF (SD V1.5, MOTION ADAPTER V3) with prompt "A futuristic park filled with holographic trees and robotic animals, with children running and playing under an artificial sky".

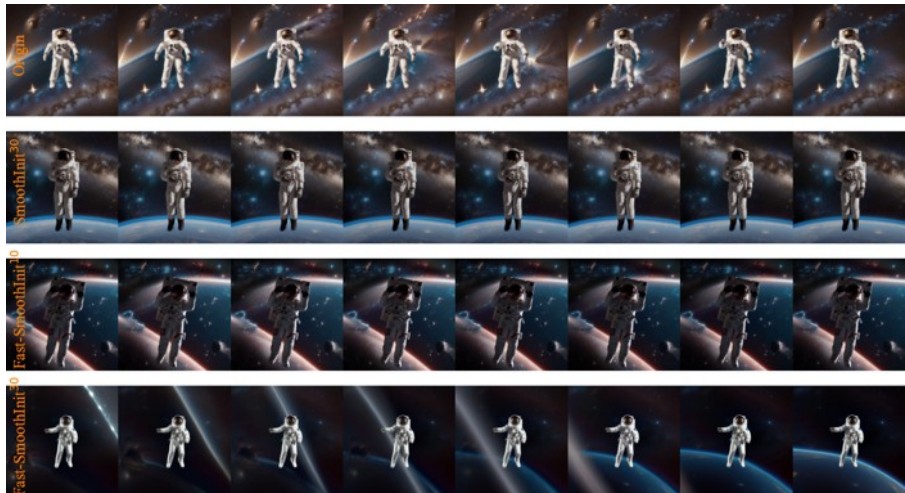

Figure 15: The synthesized video visualization of ANIMATEDIFF (SD V1.5, MOTION ADAPTER V3) with prompt "A lone astronaut floating through space, staring at the distant Earth, with stars and galaxies all around".

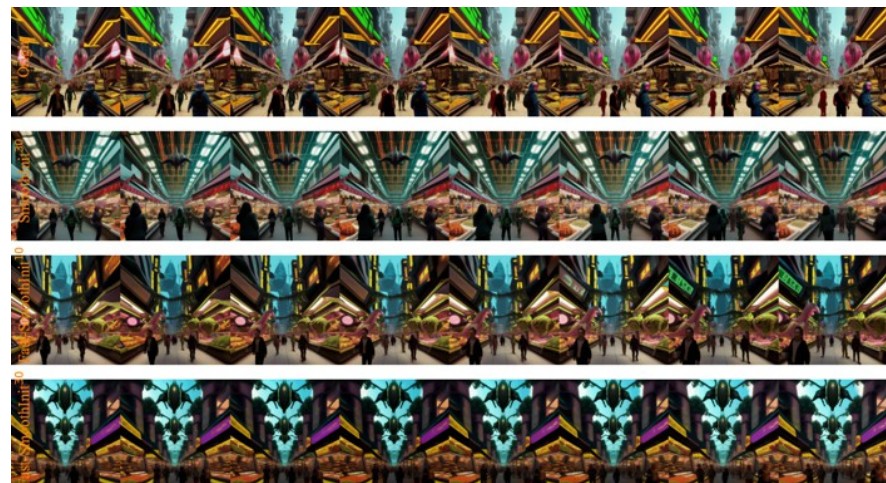

Figure 16: The synthesized video visualization of ANIMATEDIFF (SD V1.5, MOTION ADAPTER V3) with prompt "A bustling futuristic market on an alien planet, with strange creatures selling exotic goods and glowing alien plants lining the streets".

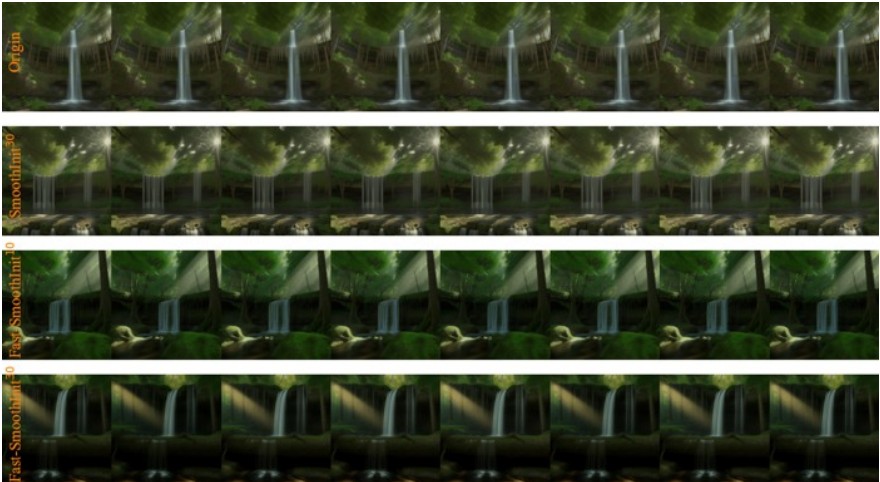

Figure 17: The synthesized video visualization of ANIMATEDIFF (SD V1.5, MOTION ADAPTER V3) with prompt "A tranquil waterfall in the middle of a dense forest, with beams of sunlight filtering through the trees and birds singing in the branches".

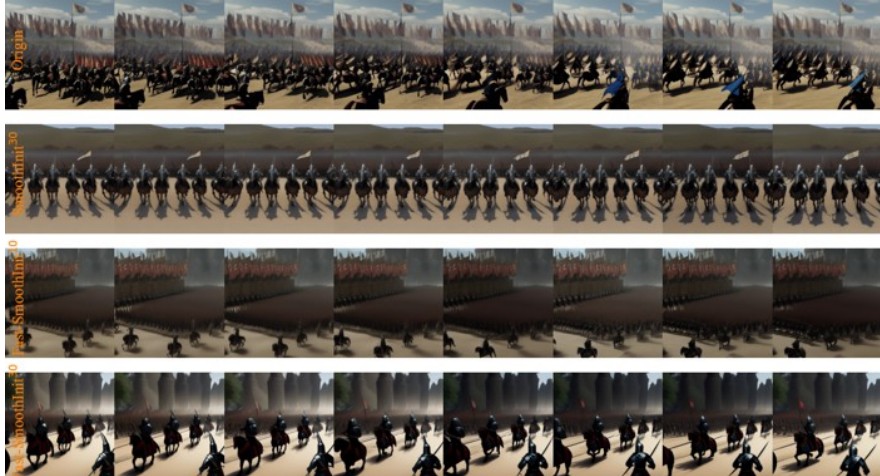

Figure 18: The synthesized video visualization of ANIMATEDIFF (SD V1.5, MOTION ADAPTER V3) with prompt "A group of knights charging into battle, their swords raised and banners flying as they face a massive army".

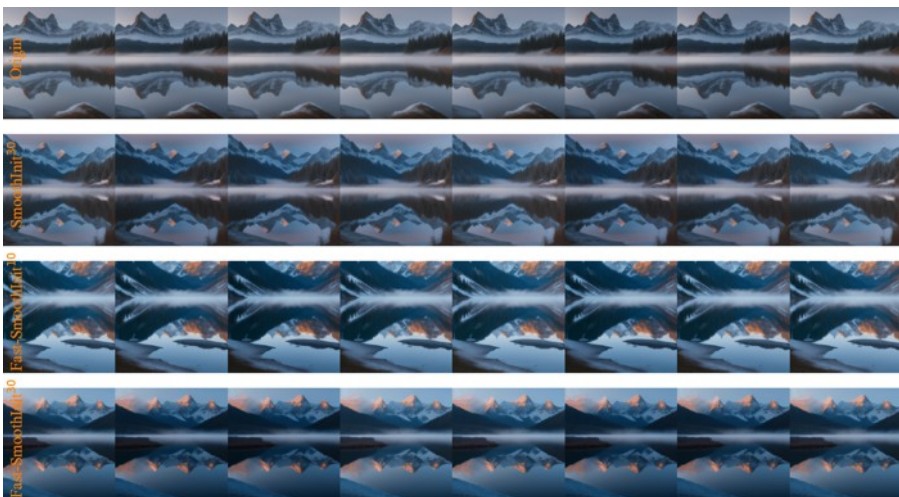

Figure 19: The synthesized video visualization of ANIMATEDIFF (SD V1.5, MOTION ADAPTER V3) with prompt "A serene mountain lake at dawn, with mist rising from the water and the reflection of snow-capped peaks mirrored on the surface".

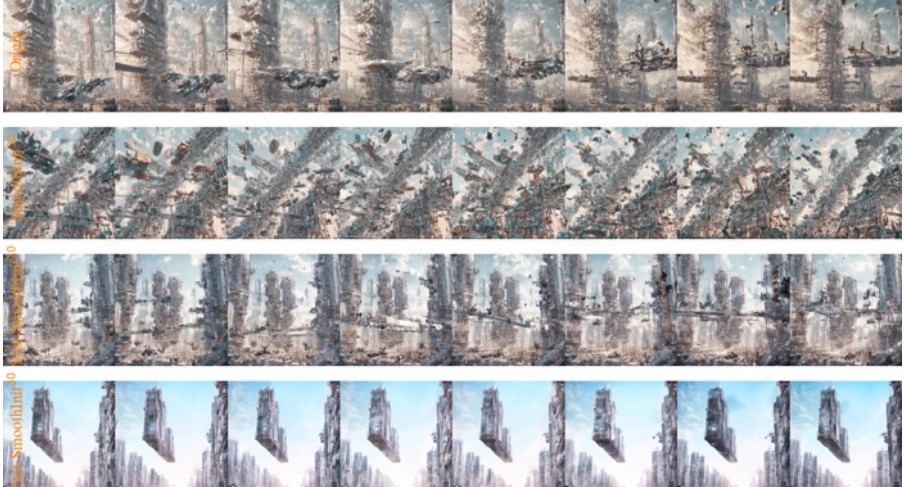

Figure 20: The synthesized video visualization of ANIMATEDIFF (SD XL, MOTION ADAPTER BETA) with prompt "A massive skyscraper under construction in a futuristic city, with robotic workers flying between the steel beams as they assemble the building".

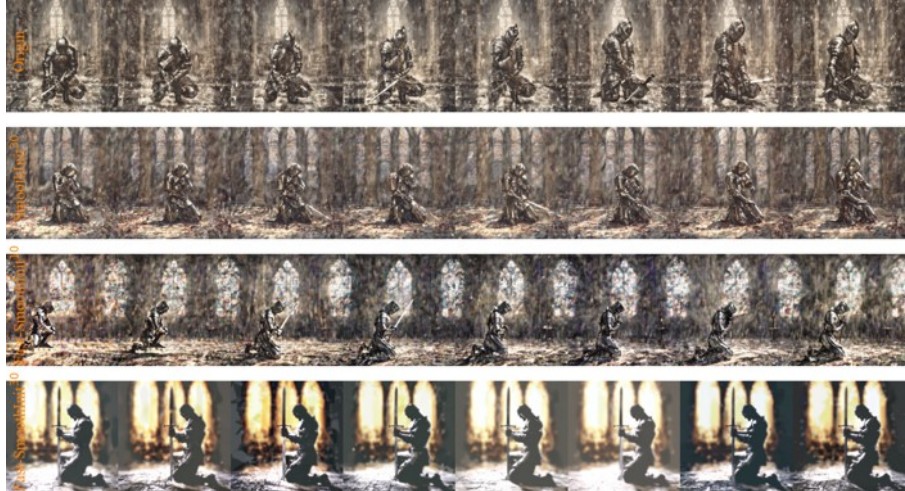

Figure 21: The synthesized video visualization of ANIMATEDIFF (SD XL, MOTION ADAPTER BETA) with prompt "A knight kneeling in a chapel, his sword laid before him as he prays for strength before a great battle".

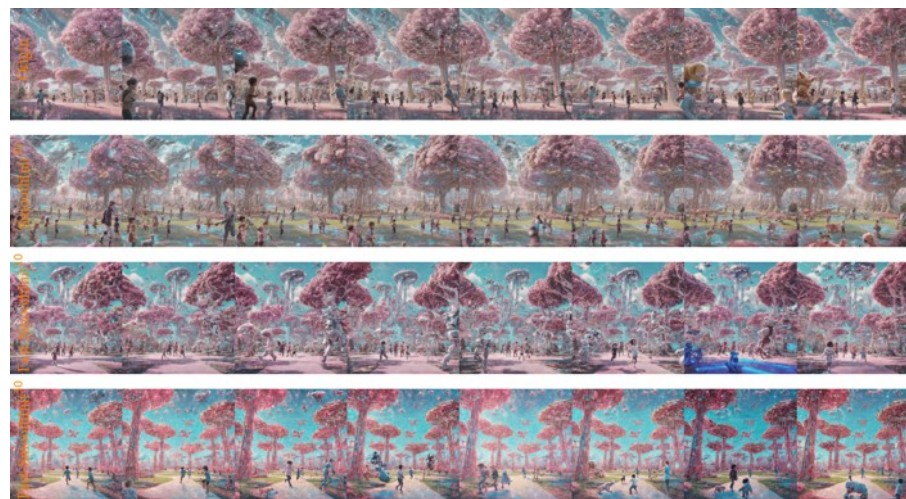

Figure 22: The synthesized video visualization of ANIMATEDIFF (SD XL, MOTION ADAPTER BETA) with prompt "A futuristic park filled with holographic trees and robotic animals, with children running and playing under an artificial sky".

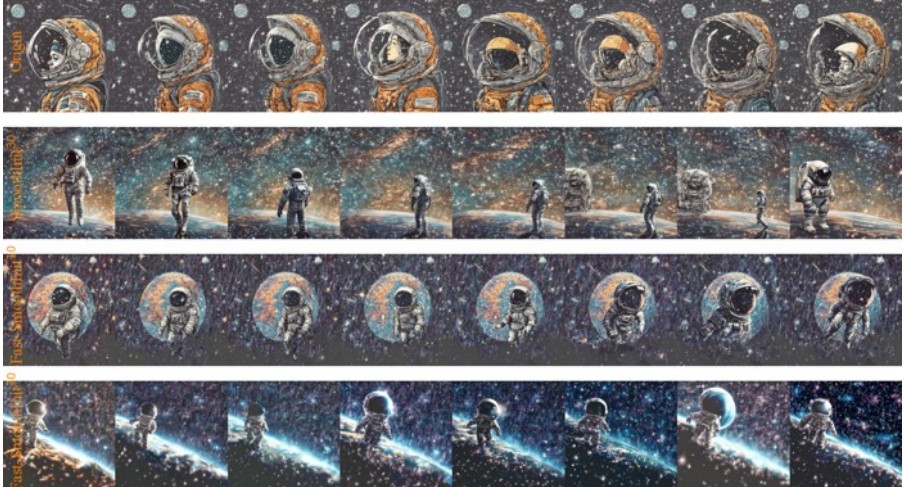

Figure 23: The synthesized video visualization of ANIMATEDIFF (SD XL, MOTION ADAPTER BETA) with prompt "A lone astronaut floating through space, staring at the distant Earth, with stars and galaxies all around".

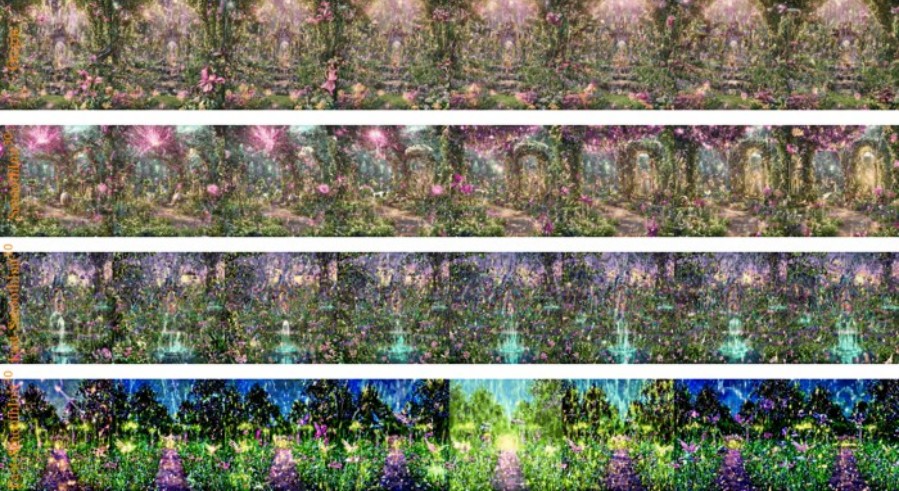

Figure 24: The synthesized video visualization of ANIMATEDIFF (SD XL, MOTION ADAPTER BETA) with prompt "A magical garden filled with glowing flowers, enchanted fountains, and mythical creatures wandering among the greenery".

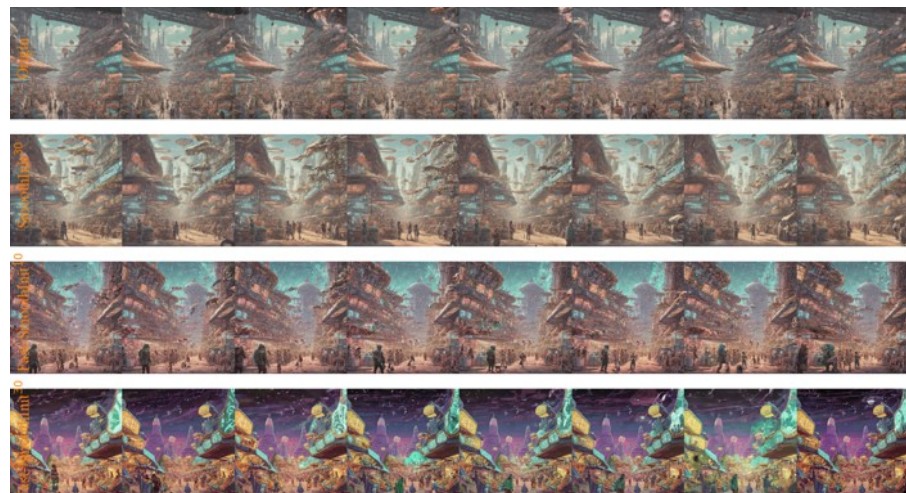

Figure 25: The synthesized video visualization of ANIMATEDIFF (SD XL, MOTION ADAPTER BETA) with prompt "A bustling futuristic market on an alien planet, with strange creatures selling exotic goods and glowing alien plants lining the streets".

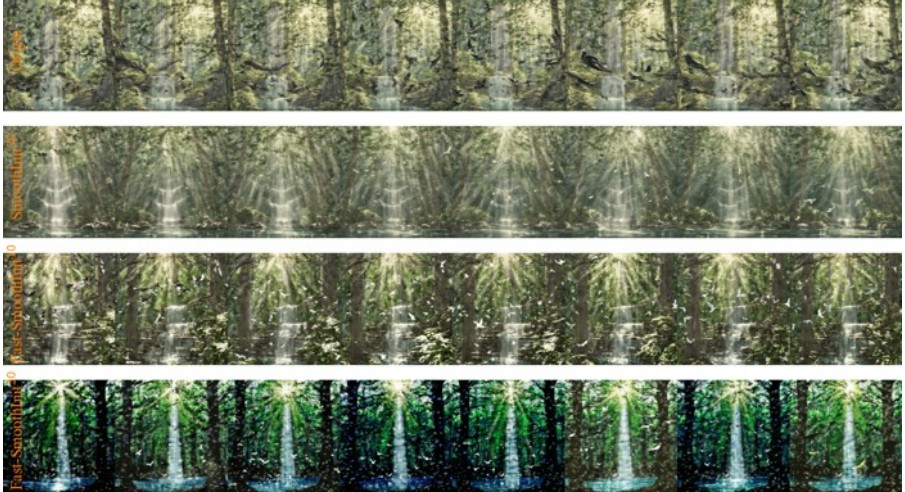

Figure 26: The synthesized video visualization of ANIMATEDIFF (SD XL, MOTION ADAPTER BETA) with prompt "A tranquil waterfall in the middle of a dense forest, with beams of sunlight filtering through the trees and birds singing in the branches".

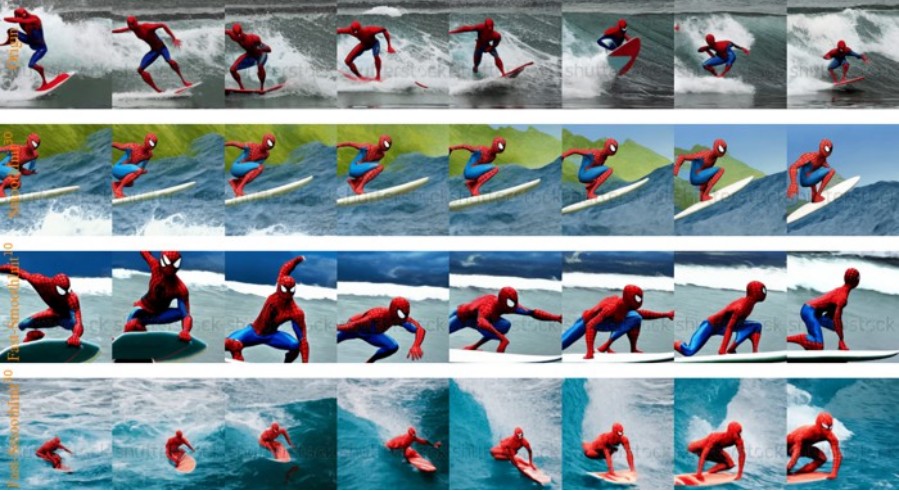

Figure 27: The synthesized video visualization of MODELSCOPE-T2V with prompt "Spiderman is surfing".

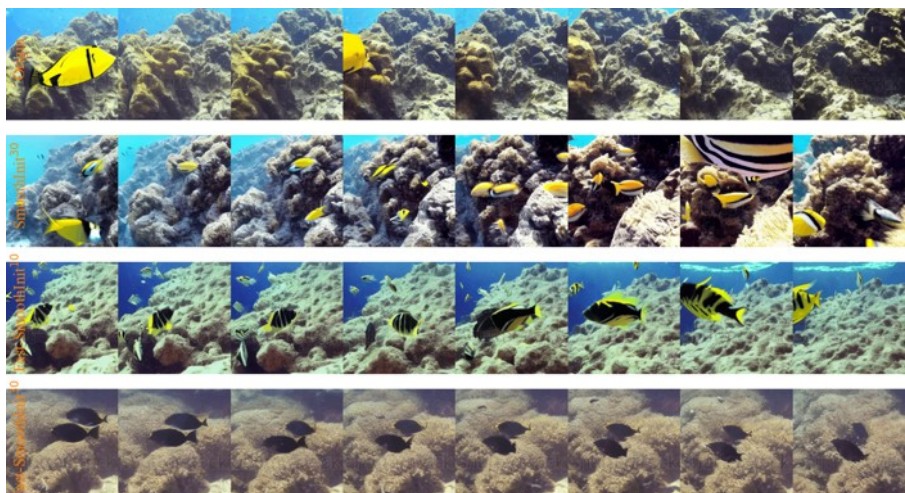

Figure 28: The synthesized video visualization of MODELSCOPE-T2V with prompt "Yellow and black tropical fish dart through the sea".

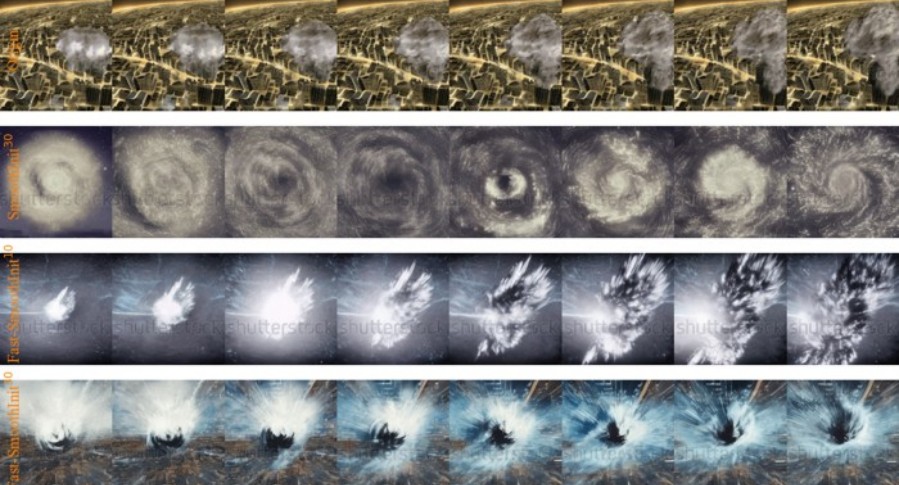

Figure 29: The synthesized video visualization of MODELSCOPE-T2V with prompt "An epic tornado attacking above a glowing city at night".

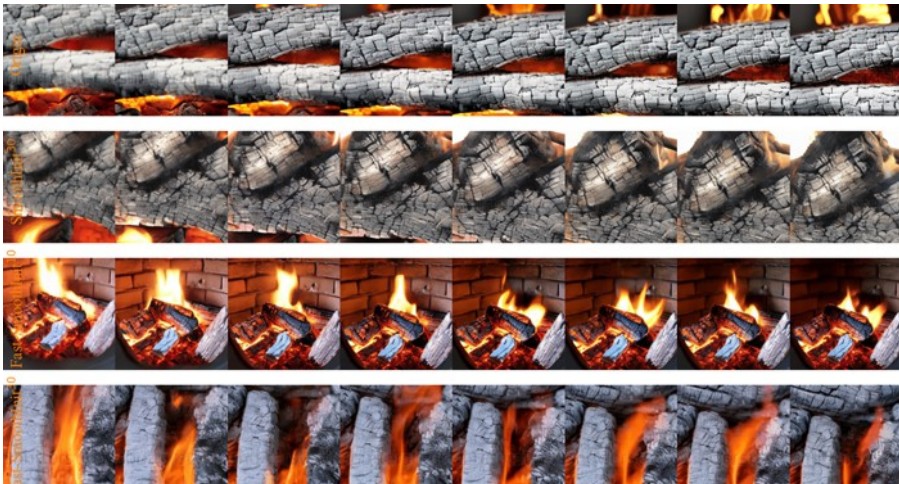

Figure 30: The synthesized video visualization of MODELSCOPE-T2V with prompt "Slow pan upward of blazing oak fire in an indoor fireplace".

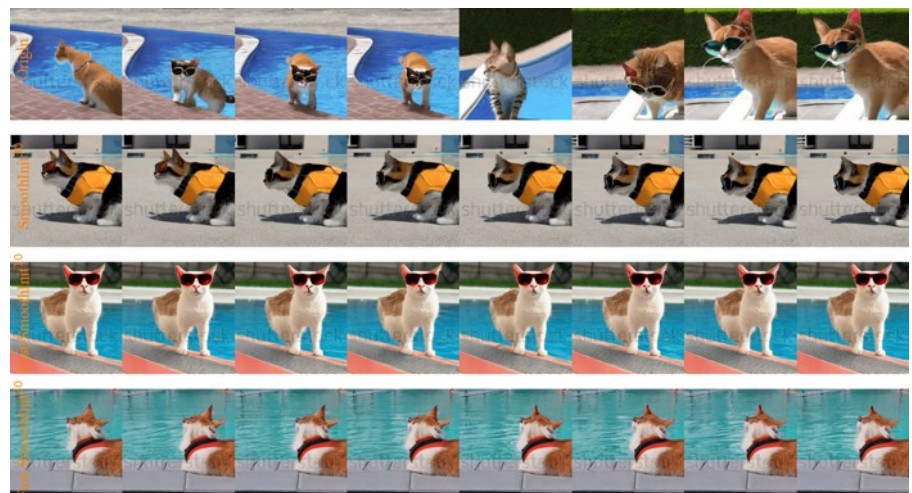

Figure 31: The synthesized video visualization of MODELSCOPE-T2V with prompt "a cat wearing sunglasses and working as a lifeguard at pool".

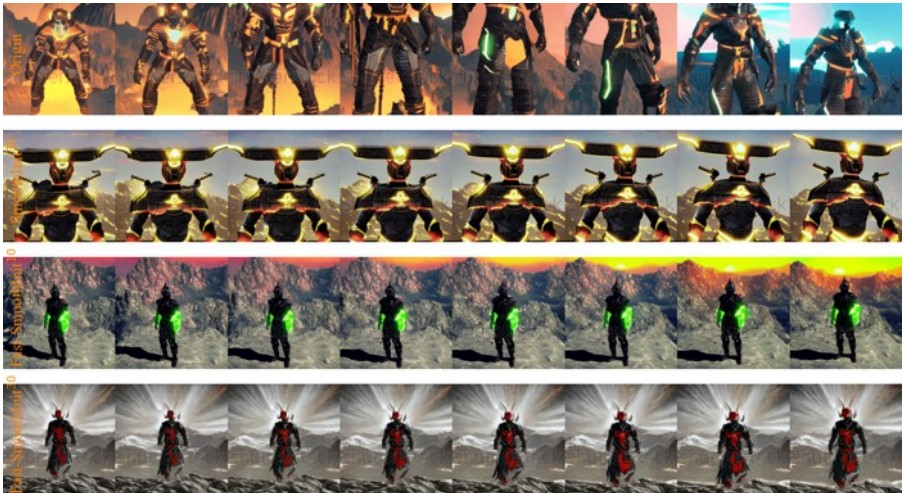

Figure 32: The synthesized video visualization of MODELSCOPE-T2V with prompt "A cybernetic samurai standing on a mountain peak, with glowing neon armor, facing a setting sun".

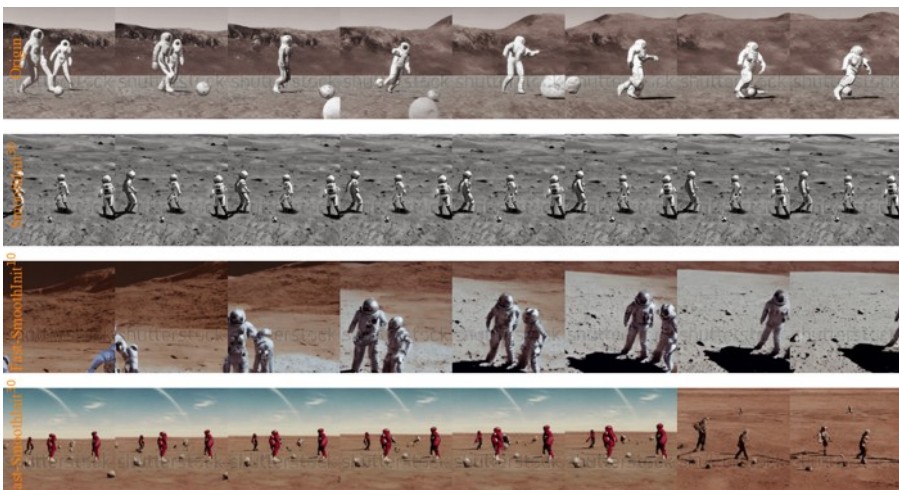

Figure 33: The synthesized video visualization of MODELSCOPE-T2V with prompt "A group of astronauts playing soccer on Mars, with the Earth visible in the background".

