# OpenReview forum: "The Blessing of Smooth Initialization for Video Diffusion Models"
_ICLR.cc/2025/Conference — ICLR 2025 Conference Withdrawn Submission_

### Official Review · Reviewer_CuyF · 2024-11-01

**Soundness:** 2
**Presentation:** 1
**Contribution:** 2
**Rating:** 3
**Confidence:** 4

**Summary:**

This paper propose a training-free paradigm which optimizes the initial Gaussian noise by introducing a targeted semantic prior bias into the sampling process from a smoothing perspective. The proposed methods Smoothinit significantly improves the fidelity and semantic faithfulness of the synthesized videos and the it has a more efficient version fast-smoothinit.

**Strengths:**

The authors provide theoretical analysis on 1. Why initial noise is important 2. Using forward-inversion could introduce future steps information into initial noises 3. The blessing of smoothing. These analyses motivate the proposed method--SmoothInit and its efficient version of fast smoothinit.

**Weaknesses:**

- While the authors present theoretical analyses of their proposed methods, the theorems as stated lack clarity and rigor with clear and correct notations and the proof processes are also not clear with some errors, leading to questions about their accuracy and validity.
  - Theorem 3.1 is not properly stated with clear defined notations. For example, $\mathcal{S}$ is a variable related with H? $\dot{\epsilon}$ is a derivative with which variable? The relation betwen $H, k, T$, etc. It is not easy to read.
Besides, in Appendix C.2, the process of how to simplify the continued product $\prod$ in (12) to keep only the main terms is not so clear. Why can just neglect the cross terms？
  - In Theorem 3.2, what is the meaning of dx/dt? It refers to the diffusion ODE functions? Seems not. Besides, in its proof, the eq. 22 integral with respect to t, where is the t? If x is related to t, then why $q_1(x_t) $and $q_{1-\eta}(x_t)$? These equations are so confusing and it even contains the basic mathematical errors.
  - Same unclear notations in the remaining theorems.

- Overhead problem. In the appendix, the authors states its overhead on the inference is less substantial compared to FREEINIT and UNICTRL but didn't provide the quantitative comparison with other methods and baseline. Besides, it should be a tradeoff between the inference overhead and quality improved. Showing this tradeoff through qualitative is important for a training-free method to validate its usuability in real applications.

- The main concern for this paper is that smooth and fast-smooth techniques are mainly for better estimation of $f_s$ where $f_s$ is an approximation DDIM-Inversion(DDIM$(\cdot))$. However, DDIM-Inversion(DDIM$(\cdot))$ alone does not guarantee that the intended future semantic information is correctly introduced into the initial noise. While DDIM-Inversion(DDIM(·)) can inject semantic information into the initial noise, it is unclear how it ensures that this information is meaningful or beneficial.

**Questions:**

- When use DDIM and DDIM inversion for editing, we usually set $w_1 = w_2=1$. Otherwise, it will amplifies the accumulated error (see [1]). In this work, the authors introduce a gap by setting  w_1 \neq w_2  in DDIM and DDIM inversion, allowing information from future steps to influence the initial noise. However, it remains unclear how they mitigate potential additional discretization errors that might affect the initial noise and, consequently, the final output.

- As DDIM and DDIM inversion are discrete ODE solvers, it would be helpful for the authors to clarify the number of DDIM and DDIM inversion steps used in their implementation, as these steps significantly impact the efficiency of the proposed method.



[1] Null-text Inversion for Editing Real Images using Guided Diffusion Models

---

> ### Author Response · Authors · 2024-11-13
> **Clarification on some points**
>
> We sincerely appreciate the feedback provided by Reviewer CuyF. Initially, we consider directly withdrawing the paper, but after carefully reviewing your first concern, we feel compelled to address it and clarify some points.
>
> **$\mathcal{S}$ is a variable related with H?**
>
> > It is intuitive that $\mathcal{S}$ is correlated with several independent variables, including H, as demonstrated in Theorem 3.1.
>
> **$\dot{\epsilon}$ a derivative with which variable?**
>
> > This is about partial derivatives of $t$, and the proof in the appendix presents this fact (going around $t$) quite intuitively.
>
> **The relation betwen $H$, $T$, $k$**
>
> > As described in Appendix C.2, $Nk = T$, where $N$ is the number of sampling steps, $k$ is the length of a single sampling step, and $T$ is the total number of sampling steps. The value of $h$ is used to derive the result defined in Theorem 3.1, which demonstrates that our theorem is applicable when sampling begins at any point.
>
> **The process of how to simplify the continued product $\prod$ in (12) to keep only the main terms is not so clear. Why can just neglect the cross terms?**
>
> > I believe this comment is incorrect. If you are familiar with the Binomial Theorem and one of the terms is $\dot{\epsilon}$, it is intuitive to me that there are exactly $N$ terms containing the first-order $\dot{\epsilon}$ in the binomial expansion.
>
> **In Theorem 3.2, what is the meaning of dx/dt? It refers to the diffusion ODE functions**
>
> > $\frac{dx}{dt}$ is naturally an ODE because it does not contain any random terms, except that the integral term in this ODE is not a function of $t$.
>
> **Besides, in its proof, the eq. 22 integral with respect to t, where is the t?**
>
> > I wanted to ask why the integral term must be the derivative with respect to $t$. It seems you have introduced the diffusion process actively, but this is not a standard diffusion process.
>
> **Same unclear notations in the remaining theorems**
>
> > I think this is subpar because all of your previous feedback is due to your own lack of math skills.
>
> **Why can just neglect the cross terms？**
>
> > The presence of the Chinese question mark "？" here seems unusual.
>
> Finally, we thank the reviewers for their suggestions and for pointing out the disadvantages in our experiments, which we will address in the next version. However, we do not agree that our mathematical derivation was problematic. We have thoroughly reviewed it multiple times and find it difficult to accept the feedback of our mathematical theorem in this manner.

---

### Official Review · Reviewer_ccA6 · 2024-11-03

**Soundness:** 2
**Presentation:** 3
**Contribution:** 2
**Rating:** 5
**Confidence:** 4

**Summary:**

This paper tackles the challenge of sub-optimal and temporally inconsistent sampling in video diffusion models by correcting the initial noise vector. The authors first emphasize the impact of the initial noise on the output by analyzing the Lipschitz constant. They then introduce a forward-inversion operator that uses varying CFG scales during the DDIM and DDIM-Inversion processes, preserving semantic information within the optimized noise. To ensure consistency, they incorporate a random smoothing process, termed SmoothInit, along with a faster variant. The authors demonstrate that this initial noise correction improves video sampling in a training-free manner.

**Strengths:**

- Theoretical contributions: The paper provides some novel theoretical insights on the lipschitzness of the diffusion models, and theoretical ground on how SmoothInit helps prevent sampling from convergent to a suboptimal solution. I think these findings can be leveraged not only for this paper but with other diffusion-based frameworks, e.g. improving and better understanding inversion, editing, etc.

- Experimental demonstrations: Authors have included large-scale ablation studies on the design space of the proposed framework, including the iteration number, noise level for smoothing, noise type, difference between SmoothInit and Fast smoothinit, etc. Also with additional smoothing, the proposed framework outperforms FreeInit which pioneers the relationship between noise initialization and video quality.

**Weaknesses:**

- Analysis Considering the Properties of Video Diffusion Models: While the proposed framework is grounded in theoretical motivations, these observations are not explicitly tailored to video diffusion models. Although this generalizability may be advantageous, more discussion is needed to explain how the proposed initialization specifically enhances sampling in the video context. For instance, FreeInit emphasizes temporal consistency in videos, noting that low-frequency components of the initial noise dictate the spatio-temporal layout of generated videos, and aims to refine these components.

- Gap Between Theory and Practice: Related to the above point, there are likely differences in applying SmoothInit to image versus video diffusion models. DDIM inversion often performs more worse in In video diffusion models then image diffusion models, potentially violating the linear approximation assumptions of the score function. Additionally, CFG scales are typically larger in video diffusion models, leading to more pronounced off-manifold noise artifacts, as shown in CFG++ [1]. Furthermore, reducing the number of inference steps is more complex for video generation compared to image generation [2], which may be linked to the failure of DDIM inversion and the curved trajectory issue. Thus, while SmoothInit in an ideal continuous setting may nudge the initial noise toward regions containing semantically rich information, in video diffusion models with time-discretized setting this process might introduce additional discrepancies, exacerbating off-manifold noise artifacts and deviating from theoretical predictions.

- Qualitative Results: The quality of the provided videos appears suboptimal, suffering from reduced motion dynamics (e.g., ModelScope S30 6), color saturation, and flickering artifacts (e.g., AnimateDiff F30 3). More comprehensive video visualizations comparing the proposed method with baselines like FreeInit would be helpful. Publishing these visualizations on an anonymous project page would further enhance accessibility and comparison.

- Quantitative Results: While the forward-inversion process improves results to some extent, its performance is only quantiatively comparable to FreeInit without smoothing. With smoothing, it outperforms other baselines, but the approach shares similarities with FreeInit, which diminishes its novelty. Additionally, it may still face the trade-off between motion dynamics and temporal consistency, a limitation also observed in FreeInit.

---
**References**

- [1] Chung, Hyungjin, et al. "CFG++: Manifold-constrained Classifier Free Guidance for Diffusion Models." arXiv preprint arXiv:2406.08070 (2024).
- [2] Polyak, Adam, et al. "Movie Gen: A Cast of Media Foundation Models." arXiv preprint arXiv:2410.13720 (2024).

**Questions:**

Overall, I appreciate the theoretical contributions and the extensive empirical demonstrations. However, there is a need for improved qualitative results and more in-depth analysis. Below is a summary of key questions related to the identified weaknesses:

- Can this initialization be applied to image diffusion models as well? If so, what are the technical advantages that video diffusion models (VDMs) gain from this initialization compared to image diffusion models?

- Are there any potential gaps between the theoretical framework and practical implementation? If so, how can these gaps be addressed or mitigated?

- Could you provide more direct visual comparisons with FreeInit or other baselines? Enhanced visual comparisons would greatly aid in assessing the method's performance, possibly through side-by-side videos or visualizations hosted on an anonymous project page.

---

### Official Review · Reviewer_EXPd · 2024-11-05

**Soundness:** 3
**Presentation:** 2
**Contribution:** 2
**Rating:** 5
**Confidence:** 3

**Summary:**

This paper starts from the observation that generated videos from video diffusion models are highly sensitive to the initial random noise when using deterministic generation methods like DDIM. The authors then propose two methods, *SmoothInit* and *Fast-SmoothInit*, that first optimize the starting noise value before performing the usual generation process. SmoothInit essentially corresponds to running a step of DDIM generation, followed by a step of DDIM inversion, while adding some additional perturbation for smoothing. Fast-SmoothInit then further improves the method through a momentum technique. The paper theoretically studies the methods and formalizes them in a differential equation framework. The authors then run various experiments on video generation benchmarks, showing favourable results compared to related work.

**Strengths:**

**Originality:** While there are related methods, such as FreeInit, the proposed methods SmoothInit and Fast-SmoothInit seem to be sufficiently novel and original, to the best of my knowledge.

**Clarity:** While it is possible to overall follow the paper, some aspects of the method remain unclear (see below in Weaknesses), which affects the paper's quality.

**Experiments:** The paper runs extensive experiments, and demonstrates that the proposed approach overall outperforms the baselines in most experiments.

**Significance and Quality:** Video generation is an important problem in generative modeling that has recently gotten a lot of attention. The sensitivity of video diffusion models to their initial noise value is a problem that affects performance, and addressing this is a meaningful endeavour. The paper achieves this, as validated in its experiments. Hence, I consider the work sufficiently significant. Due to the mix of strengths and weaknesses and some issues in the paper's presentation, I consider the paper of average quality.

**Weaknesses:**

There are also various weaknesses.

- The paper relies on dense math in several places, which is sometimes hard to follow. For instance, the *smoothing* theory will not be known to most readers and I think most readers, myself included, will not be able to intuitively understand highly technical theorems like Theorem 3.3. I would suggest the authors to explain and present the intuitions here (as well as in the smoothing paragraph on page 3) better. The concrete value of these theorems remains somewhat unclear. This should be explained better and supported with more intuitions and maybe examples.
- Somewhat related, I believe the authors should better explain some obvious questions around the proposed method. Why is it that generation is so sensitive to the initial noise in video models, but less so in image models? What is special about video generation? Algorithmically, it's the same process. This should be discussed.
- When optimizing the noise with the proposed methods, how does the latent evolve? Do noise values for different frames become more similar? Does this make the initialization maybe better? This should be analyzed and discussed, too (and could give a hint why the situation is different for video models, compared to image model). The paper essentially proposes a new method that seems to empirically work but does not sufficiently analyze the why and how.
- The paper should also better put the work in context with respect to related work, in particular FreeInit (I am aware of section D in the appendix, but this is just a high-level summary of related work). FreeInit refines the spatial-temporal low-frequency components of the initial latent. The proposed methods here instead rely on a *smoothing* framework. It is known that smoothing also corresponds to a removal of low frequency information. Hence, is there maybe a deeper relation here between the methods? Does the smoothing in this paper somehow adjust low frequency components of the initial noise, too? I am just speculating here, but I think these relations and questions should could be discussed and maybe analyzed more.
- Experimentally, some of the results are somewhat incremental and some results are so close together that it remains unclear how statistically significant they are. This could be addressed by estimating errors and also reporting error bars.
- How are the y-axes of Figure 6, bottom, created? Are these L2 distances, all independently normalized to be in [0,1] for each plot? It is not clear how comparable these plots are. Related, in section 4.4, the authors write ''Specifically, at the endpoint of the sampling process (timestep = 1 in Fig.6), the video generated by Fast-SmoothInit tends to be more uniformly distributed within a sphere, whereas
SmoothInit, Forward-Inversion, and Origin tend to be distributed along a spherical shell''. This is not clear, unless I am misunderstanding something. Also for SmoothInit, trajectories spread almost the entire [0,1] interval at t = 1. This should be discussed better.

**Conclusion:** While the paper proposes a novel method and shows favourable empirical results, there remain many questions. Therefore, I don't think the paper meets the bar for acceptance in its current form, but I would be open to raising my score if my concerns and questions can be addressed.

**Questions:**

I have asked most questions already above. Just one more: the paper leverages deterministic DDIM sampling to generate the output after optimizing the noise. How does the method compare to stochastic DDPM sampling?

---

### Author Response · Authors · 2024-11-13

Dear Reviewers

We thank you for your dedication and valuable suggestions. After careful consideration, we have decided to withdraw the manuscript. In future versions, we will refine both the presentation and address the methodological and experimental deficiencies.

Best,
Authors

---

### Note · Authors · 2024-11-13

I have read and agree with the venue's withdrawal policy on behalf of myself and my co-authors.